# Endolysosomal trafficking controls yolk granule biogenesis in vitellogenic *Drosophila* oocytes

Yue Yu[1,2☯], Dongsheng Chen[1,3☯], Stephen M. Farmer[1,2,4☯], Shiyu Xu[1], Beatriz Rios[1,2], Amanda Solbach[1,2,5], Xin Ye[1], Lili Ye[1], Sheng Zhang[1,2,5,6]*

1 The Brown Foundation Institute of Molecular Medicine, McGovern Medical School at the University of Texas Health Science Center at Houston (UTHealth), Houston, Texas, United States of America, 2 Program in Neuroscience, The University of Texas MD Anderson Cancer Center UTHealth Graduate School of Biomedical Sciences (MD Anderson UTHealth GSBS), Houston, Texas, United States of America, 3 The College of Life Sciences, Anhui Normal University, #1 Beijing East Road, Wuhu, Anhui, People's Republic of China, 4 Program in Biochemistry and Cell Biology, The University of Texas MD Anderson Cancer Center UTHealth Graduate School of Biomedical Sciences (MD Anderson UTHealth GSBS), Houston, Texas, United States of America, 5 Programs in Genetics and Epigenetics, The University of Texas MD Anderson Cancer Center UTHealth Graduate School of Biomedical Sciences (MD Anderson UTHealth GSBS), Houston, Texas, United States of America, 6 Department of Neurobiology and Anatomy, McGovern Medical School at the University of Texas Health Science Center at Houston (UTHealth), Houston, Texas, United States of America

☯ These authors contributed equally to this work.
* Sheng.Zhang@uth.tmc.edu

**Data Availability Statement:** All data are in the manuscript and its supporting information files.

**Funding:** This work is supported by National Institutes of Health (NIH) grant R01 NS110943 (to S.Z.). The funders had no role in study design, data

## Abstract

Endocytosis and endolysosomal trafficking are essential for almost all aspects of physiological functions of eukaryotic cells. As our understanding on these membrane trafficking events are mostly from studies in yeast and cultured mammalian cells, one challenge is to systematically evaluate the findings from these cell-based studies in multicellular organisms under physiological settings. One potentially valuable *in vivo* system to address this challenge is the vitellogenic oocyte in *Drosophila*, which undergoes extensive endocytosis by Yolkless (Yl), a low-density lipoprotein receptor (LDLR), to uptake extracellular lipoproteins into oocytes and package them into a specialized lysosome, the yolk granule, for storage and usage during later development. However, by now there is still a lack of sufficient understanding on the molecular and cellular processes that control yolk granule biogenesis. Here, by creating genome-tagging lines for Yl receptor and analyzing its distribution in vitellogenic oocytes, we observed a close association of different endosomal structures with distinct phosphoinositides and actin cytoskeleton dynamics. We further showed that Rab5 and Rab11, but surprisingly not Rab4 and Rab7, are essential for yolk granules biogenesis. Instead, we uncovered evidence for a potential role of Rab7 in actin regulation and observed a notable overlap of Rab4 and Rab7, two Rab GTPases that have long been proposed to have distinct spatial distribution and functional roles during endolysosomal trafficking. Through a small-scale RNA interference (RNAi) screen on a set of reported Rab5 effectors, we showed that yolk granule biogenesis largely follows the canonical endolysosomal trafficking and maturation processes. Further, the data suggest that the RAVE/V-ATPase complexes function upstream of or in parallel with Rab7, and are involved in earlier stages of

collection and analysis, decision to publish, or preparation of the manuscript.

**Competing interests:** We have read the journal's policy and the authors of this manuscript have no competing interests. All research were conceptualized, planned, and conducted at the University of Texas Health Science Center at Houston (UTHealth).

endosomal trafficking events. Together, our study provides s novel insights into endolysosomal pathways and establishes vitellogenic oocyte in *Drosophila* as an excellent *in vivo* model for dissecting the highly complex membrane trafficking events in metazoan.

## Author summary

Endocytosis and endolysosomal trafficking are membrane-based package and delivery systems in eukaryotes for material exchanges intracellularly in-between membrane-enclosed compartments and extracellular with surrounding milieu. Current understanding of these exchange mechanisms are mainly from studies in yeast and cultured mammalian cells, but exactly how they operate in multicellular organisms under physiological conditions remain unclear. Here we focus on vitellogenic oocytes in *Drosophila*, which uptake large quantity of extracellular lipoproteins by a low-density lipoprotein receptor called Yolkless into the oocyte and package them into large yolk granules, a specialized lysosome, for storage and usage in later development. Using novel fly lines that allows faithful detection and manipulation of endogenous Yolkless receptor and known endosomal regulators, we show that the formation and maturation of yolk granules largely follows the canonical endolysosomal trafficking pathways, including the critical involvement of small GTPases Rab5 and Rab11 as well as distinct phospholipid species and actin networks, although the results also raise questions on the roles of other regulators including Rab4 and Rab7 in granule biogenesis. Together, this study provides novel insights into the highly complex membrane trafficking events in multicellular organisms and supports *Drosophila* oocyte as a useful *in vivo* model for such studies in the future.

## Introduction

Endocytosis and the ensuing cascade of endolysosomal trafficking are membrane-based mechanisms to interact with extracellular environment and coordinate intracellular maintenance and responses [1–5]. Essential for almost all physiological functions of eukaryotes from signaling to morphogenesis and defense, their dysfunctions are being linked to a growing number of diseases from cancer to ageing and neurodegeneration [6–13]. In case of the clathrin-dependent endocytosis, a main pathway for receptor mediated endocytosis, the cascade is a continuous and highly dynamic vesicular events involving cargo recognition and sequestration by specific receptors, formation of clathrin-coated pits (CCPs), followed by membrane fission that generates clathrin coated vesicles (CCVs) [11]. After de-coating to shed the clathrin and associated factors from CCVs, the naked endocytic vesicles fuse with each other and with early endosomes through heterotypic and homotypic fusion to grow while being transported along cytoskeleton network, gradually maturate into late endosomes accompanied by characteristic morphological changes, and eventually fuse with lysosomes for the final degradation of enclosed cargoes [4,5]. This sequential maturation process is marked by progressive modification of the external membrane composition, and gradually acidification of internal endosomal lumen that promotes the separation of cargoes from their cognate receptors followed by their active sorting into discrete degradative or recycling routes. Such a highly dynamic and heterogeneous process is orchestrated in a spatially and temporally controlled manner by a group of small Rab GTPases, including Rab5 that controls early endocytosis, Rab4 and Rab11 that mediate fast- and slow-recycling pathways, respectively, and Rab7 that promotes endosomal

maturation into late endosome and their transportation and eventual fusion with the lysosome [14–19]. For example, as the master regulator of early endocytosis, active Rab5 on endocytic vesicles and early endosomes recruits a diverse set of downstream effectors, including phosphatases for the turnover of plasma membrane-enriched phosphatidylinositol (4,5)-bisphosphate (PI(4,5)P2), the class III PI3 kinase complex to locally produce phosphatidylinositol (3)-phosphate (PI(3)P) that acts as a landmark for early endosomes to enlist additional effectors such as EEA1 and Hrs that promote endosomal tethering, fusion, intralumenal vesicle formation and trafficking [4,20–26]. CCZ1/Mon1 complex, which has been shown to interact with both active Rab5 and PI(3)P, in turn acts as a guanine nucleotide exchange factor (GEF) to recruit Rab7 onto early endosomes to promote their maturation into late endosomes [27,28]. Thus, endocytosis and endosomal maturation are characterized by the gradual conversions of phosphatidylinositol (PI) species and associated Rab proteins, from Rab5 on endocytic vesicles and early endosomes to Rab7 on late endosomes, and from plasma membrane PI(4,5) P2 to PI(3)P on early endosomes as its identity marker [4,14,19,22].

Most of our understanding of these complex membrane trafficking events are from studies in yeast and cultured mammalian cells, but how they operate in multicellular organisms under physiological setting remains largely unknown, as the significant heterogeneity of endosome morphology and miniature sizes of endosomal intermediates further complicate such studies [29]. Just as clathrin-dependent endocytosis was first observed in the oocyte of insect mosquito Aedes *aegypti L* [30], one potentially valuable metazoan model for studing endosomal trafficking is vitellogenic stage oocyte in *Drosophila*, a classic model organism that has been invaluable in discovering the conserved signaling pathways and dissecting the myriad of developmental and cellular processes in metazoans, including endocytosis and endolysosomal trafficking [31–33]. Newly deposited fly eggs are filled with large yolk granules, which are specialized storage lysosomes containing densely packed vitellogenins (yolk proteins) to support later embryogenesis [34–37]. Vitellogenins are lipoproteins synthesized exclusively outside the oocyte, by fat bodies and surrounding follicle cells [38–41]. The biogenesis of yolk granules is controlled solely by receptor-mediated endocytosis [42–44]. Vitellogenesis occurs during stages 8–10 of egg chamber development, when oocytes undergo a dramatic, more than 200-fold increase in volume within a ~16 hour window [45,46], primarily due to the massive uptake of yolk proteins from circulating hemolymphs into the oocytes by Yolkless (Yl), a low-density lipoprotein receptor (LDLR) for Yolk proteins [47,48]. After Yl-mediated internalization of Yolk proteins and their dissociation, Yl is quickly recycled back to the plasma membrane for numerous more rounds of endocytosis [48–51]. The final mature granules are large spheres with highly acidic lumen packed with crystalized yolk proteins surrounded by a single layer of limiting membrane with characteristic smooth surface and rounded shape [52–57]. Importantly, as the vitellogenic stage oocyte is a single cell of huge sizes and dedicated almost exclusively to endocytosis, it is filled with large number of easily detectable endosomal intermediates of different maturation stages, many with unusually large sizes [52–57]. Therefore, even a mild disruption of an endosomal trafficking step can have a dramatic amplifying effect, leading to easily identifiable phenotypes, which make the oocyte a highly sensitive *in vivo* model to study receptor-mediated endocytosis and endosomal trafficking.

Despite these unique features, there are few systematic studies on the molecular and cellular mechanisms that control yolk granule biogenesis in *Drosophila*. In this study, to facilitate the analyses of receptor-mediated endocytosis in oocytes, we created genome-tagging fly lines that express a functional Yl receptor with HA and fluorescent eGFP tags. We next analyzed the expression patterns and functional requirements of four major Rab GTPases as well as PI(4,5) P2 and PI(3)P phosphoinositides in vitellogenic oocytes. Focusing on a set of Rab5 effectors identified in a recent proteomic study [58], we carried out RNA interference (RNAi) studies to interrogate their roles in yolk granule biogenesis. Our results provide new insights into

endocytosis and endolysosomal trafficking processes under this physiological setting and demonstrate vitellogenic oocyte as an excellent *in vivo* model for dissecting these highly complex yet important biological processes.

## Results

### Dynamic Yl distribution in vitellogenesis oocytes

To minimize artifacts often associated with ectopically expressed transgenes, we engineered a Yl genome-tagging construct by cloning a ~12kb genomic DNA that contains all the introns and exons as well as 5' and 3' UTR regions of the native *yl* gene together with an in-frame insertion of eGFP and 3 copies of HA epitope (eGFP-3xHA) near the C-terminal short cytoplasmic tail of encoded Yl receptor, just in front of one of the predicted endocytic signal sequences (Fig 1A and 1B) [47]. The ~5.4kb 5' UTR within the construct included the known regulatory region of *yl* gene that controls its specific expression in female oocytes [47]. Western blot analyses confirmed the expression of Yl-eGFP-3xHA fusion at the predicted ~240 kDa size [48] only in female transgenic flies (Fig 1C). Immunofluorescent staining and confocal analyses further demonstrated that the Yl-eGFP-3xHA fusion is expressed in a pattern and with characteristic subcellular distribution similar to that of endogenous Yl (Fig 1D–1G) [48]. For example, in stage 8 egg chamber when vitellogenesis started, Yl-eGFP-3xHA became mobilized from intracellular granules to the plasma membrane to initiate the extracellular upake of Yolk proteins, as revealed by their co-localization on small puncta along the cortex and on small vesicular structures inside the oocyte (insets in Fig 1D). Further interior, Yl-eGFP-3xHA was largely absent from larger Yolk-positive granules (Fig 1D). By stage 10, when vitellogenesis reaches its peak and the oocyte becomes engaged in intensive Yl-mediated Yolk uptake, Yl-eGFP-3xHA showed an almost exclusive localization to the plasma membrane (Fig 1E). Closer inspection revealed numerous Yl-positive puncta, likely representing individual endocytosis units, on the surface of the plasma membrane (Fig 1F and 1G). This is consistent with the ultra-structures revealed by transmission electron microscope (TEM) (Fig 1H–1J) [46,55], which showed the presence of extensive invaginating CCPs and CCVs on or near the plasma membrane (green arrows in Fig 1J) and larger, irregular shaped tubular and vesicular structures nearby that contained electron-dense materials immersed in the middle of electron-lucent spaces, which likely corresponded to developing endosomes (red arrows in Fig 1I and 1J). Further interior were progressively larger and spherically shaped yolk spheres with irregular boundaries that were filled with electron-dense materials but largely devoid of electron-lucent materials, which likely corresponded to growing late endosomes as they reach their final maturation stages (labeled with "i" for immature granules in Fig 1I). The mature yolk granules were easily recognized for their characteristic spherical shape, smooth membrane boundary and darker electron-dense core with crystalized Yolk protein condensates (labeled with "m" for mature granules in Fig 1I) [57]. Importantly, homozygous adults for *yl*[13], a strong loss of function allele (also known as *k621*) of *yl* gene, were females sterile [44,59]. In the presence of Yl-eGFP-3xHA transgene, this sterile phenotype was fully rescued and the homozygous *yl*[13] flies can be maintained as stand-alone stock. Together, these findings support that the genome-tagged Yl-eGFP-3xHA is functional and trafficked similarly as endogenous Yl. We thus used this Yl genome-tagging construct in subsequent studies.

### Expression and subcellular distribution of Rab4, 5, 7 and 11 in stage 10 oocytes

Among all the Rab GTPase proteins, Rab4, Rab5, Rab7, and Rab11 are known for their well-established roles in different steps of endosomal trafficking. However, except for Rab5, which

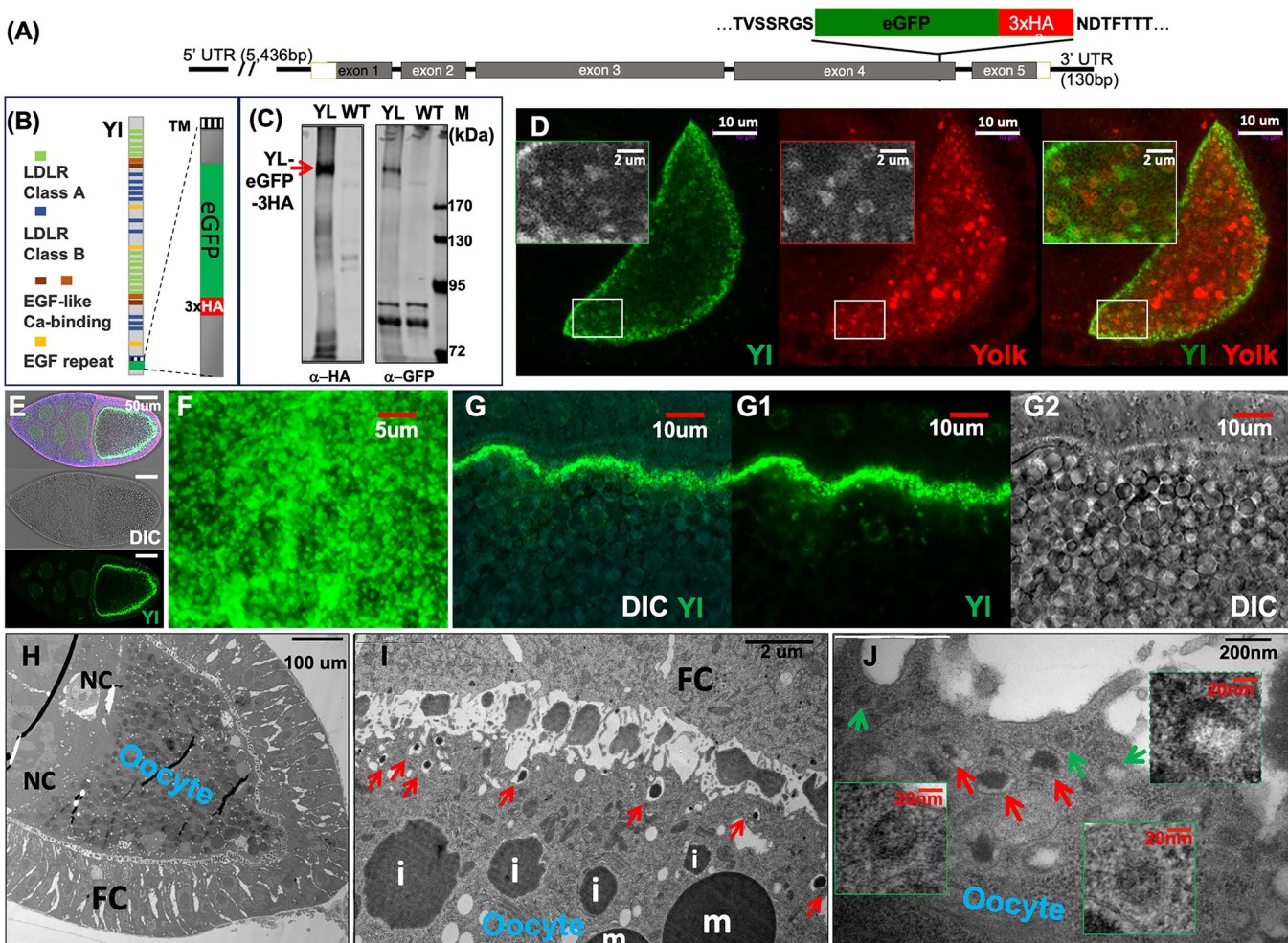

**Fig 1. Genome-tagging of Yl receptor and Yl-mediated Yolk endocytosis in vitellogenic oocyte.** (A) Schematics of Yl genome-tagging construct. The amino acid sequences of the encoded Yl protein that flank the eGFP-3xHA insertion is annotated on top of the illustration. Black lines: introns and 5' and 3' untranslated regions (UTR) of *yl* gene. Rectangular boxes: exons of *yl* gene with the coding regions shaded in gray. (B) Schematics of Yl-eGFP-3xHA fusion protein (left) and closeup view of the C-terminal cytoplasmic tail of Yl protein with the eGFP-3xHA insertion (right). TM, transmembrane domain. (C) Western blot analysis of homogenates from adult female flies probed independently with anti-HA or anti-GFP antibodies, as indicated. A single ~240kDa band positive for both GFP and HA was present in the flies transgenic for Yl-eGFP-3xHA genome-tagging construct (YL) but absent in $w^{1118}$ non-transgenic (WT) control. (D) Stage 8 oocyte double-labeled by anti-GFP for Yl-eGFP-3xHA (green) and anti-Yolk (red) proteins, as annotated. Inserts are zoom-in view of the region highlighted in white box, showing enrichment of Yl along the oocyte cortex and the differential distribution of Yl and Yolk on granules of different sizes inside the oocyte. (E-G) Expression and subcellular localization of Yl-eGFP-3xHA in stage 10 oocytes imaged for Yl-GFP-3xHA by anti-GFP (green) and yolk granules (gray) by differential interference contrast (DIC). (E) Low-magnification view of whole egg chamber, showing (bottom) the localization of Yl along the oocyte plasma membrane, (middle) yolk granules inside the oocytes and (top) their overlaying view. (F, G) Surface (F) and cross-section (G) views of a plasma membrane area at high-magnifications. (H-J) TEM of stage 9 egg chambers. (H) Low-magnifications overview, showing an oocyte filled with many electron-dense yolk granules surrounded by follicle cells (FC) and nurse cells (NC), as annotated. (I) Closer-up view of the cortex region of an oocyte. Notice that along the plasma membrane, the presence of many small granules (red arrows) with characteristic electron-dense core in the middle of translucent space surrounded by a single layer of membrane, larger granules of growing sizes with irregular boundaries (marked with "i") and large mature granules (marked with "m"). (J) High-magnification view of plasma membrane area, showing the presence of CCPs and CCVs (green arrows) and nearby early endosomes (red arrows). Genotypes for (D-G): $w^{1118}$; p{mini-W+, *yl*-eGFP-3xHA}. (H-J) $w^{1118}$. The sizes of the scale bars as annotated inside images.

was shown to be essential for Yl-mediated endocytosis and granule biogenesis [49], the expression and functions of the other Rab proteins in vitellogenic oocytes have not been fully examined in detail. Focusing on wildtype stage 10 oocytes, when endocytosis reaches its peak, we examined the expression and subcellular distribution of these Rab proteins, using Yl as an endocytosis marker. Yl showed a significant co-localization with Rab7 along the oocyte cortex,

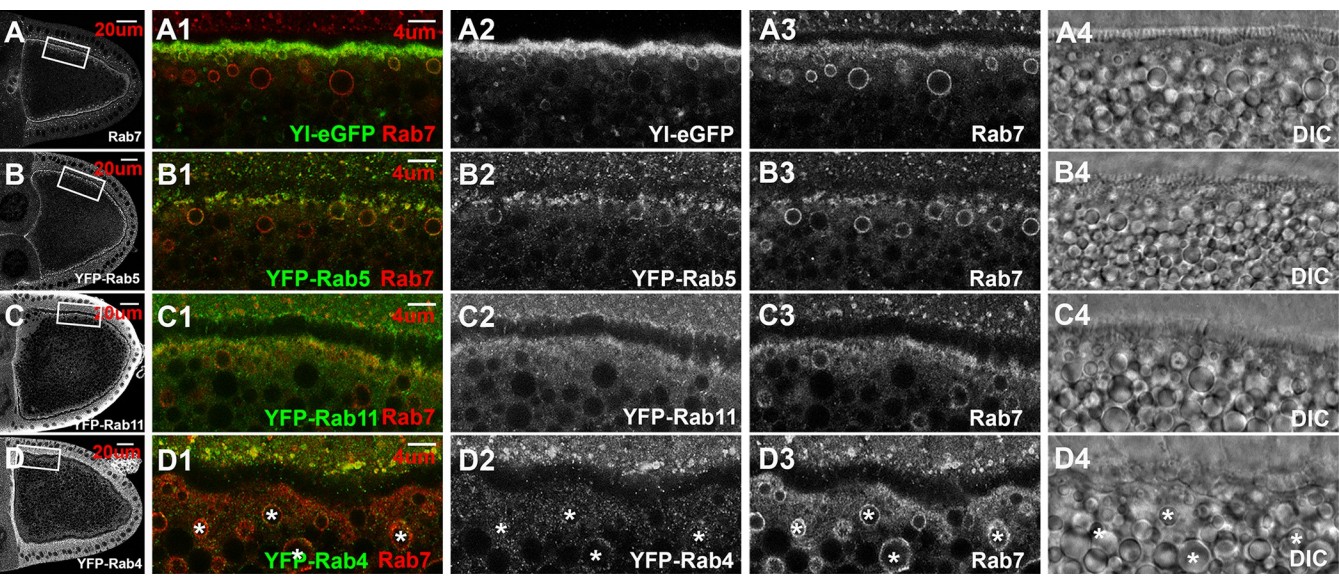

**Fig 2. Expression and subcellular localization of Rab4, Rab5, Rab7 and Rab11 in vitellogenic oocytes.** Confocal images of stage 10 egg chambers double-labeled for endogenous Rab7 (red) and eGFP or YFP tag (green) for (A) Yl-eGFP-3xHA, (B) YFP-Rab5, (C) YFP-Rab11 and (D) YFP-Rab4, respectively, from the corresponding endogenous tagging fly lines. Yolk granules within the same oocytes were visualized by DIC (gray). (A-D) Overview of Rab7 (A), YFP-Rab5 (B), YFP-Rab11 (C) and YFP-Rab4 (D) in the imaged oocytes, respectively. (A1-D4) High-magnification view of the corresponding cortex regions highlighted in (A-D), respectively, as indicated. Endogenous Rab7 used as reference for endosomes in all the images. Images shown in individual channels in gray or as overlaying images in color, as indicated. Genotypes: (A) $w^{1118}$; p{w(+mC), *yl*-eGFP-3xHA}. (B) $w^{1118}$; TI{TI} EYFP-Rab5 /CyO (BDSC #62543). (C) $w^{1118}$; TI {TI} EYFP-Rab11 (BDSC #62549). (D). $y^1$, $w^{1118}$; TI{TI} EYFP-Rab4 (BDSC #62542). The sizes of the scale bars as labeled.

as revealed by co-staining with a specific anti-Rab7 antibody [60] (Fig 2A). The two also partially overlapped on small vesicles immediately adjacent to the plasma membrane (Fig 2A1–2A3). A little further inside, larger granules strongly positive for Rab7 but negative for Yl were present. Rab7 was notably absent on large granules that located further interior, indicating a tight spatial regulation of Rab7 recruitment along the granule biogenesis route. Interestingly, co-labeling with lysotracker, a marker for highly acidic cellular compartments, revealed that in the oocyte cortex, small-sized lysotracker-positive granules, but not the larger ones, were often surrounded by weak YL-GFP signal (S7A–S7C Fig).

In oocytes from YFP-Rab5 endogenous tagging line [61], YFP-Rab5 showed a predominant enrichment on the cortex, similar as that reported in a previous study using an anti-Rab5 antibody (Fig 2B) [49]. YFP-Rab5 partially co-localized with Rab7 on puncta-like structures along the cortex (Fig 2B1). Similar to Yl, YFP-Rab5 also co-localized with Rab7 on granules immediately adjacent to the plasma membrane, but its signal faded rapidly on Rab7-positive granules located further interior (Fig 2B1–2B3). Noticeably, on granules where both Rab5 and Rab7 co-existed, the two showed an uneven and non-identical distribution along the granule membrane, consistent with the models of Rab subdomains and Rab5-to-Rab7 conversion that govern endosomal maturation [17,18,25].

Examination in both YFP-Rab11 endogenous tagging line [61] and by an anti-Rab11 antibody revealed similar enrichment of Rab11 along the whole cortex region in stage 10 oocytes (Figs 2C and S1C–S1F), a pattern that is different from a previous report showing a specific localization of Rab11 at the extreme posterior pole of the oocyte [62]. Unlike Rab5 and Rab7, Rab11 showed no clear association with vesicular structures (Figs 2C1–2C3 and S1C–S1F).

Surprisingly, despite its well-documented role in fast recycling, in YFP-Rab4 endogenous tagging flies [61], no clear enrichment of YFP-Rab4 was observed along the oocyte cortex. Instead, YFP-Rab4 showed a weak and scattered distribution as puncta inside the oocytes

(Fig 2D) and intriguingly, partially co-localized with Rab7-positive granules (highlighted by asterisks in Fig 2D1–2D4).

Importantly, GFP signals from the YFP-Rab5, -Rab11 and -Rab4 genome tagging lines were specific, as when processed in parallel with the same anti-GFP antibody, only low background signals were detected in controls that did not carry YFP-tagging lines (S1A–S1D Fig).

## Actin cytoskeleton dynamics and phosphatidylinositols in vitellogenic oocytes

Considering the importance of actin dynamics in endocytosis and endosomal trafficking [63], we next examined the distribution of actin cytoskeleton in vitellogenic oocytes. In stage 10 oocytes, phalloidin staining revealed a dense layer of filamentous actin (F-Actin) that largely overlapped with the Rab7-positive layer and the top half of Yl-positive band along the plasma membrane (Fig 3A and 3B). Additionally, a thin web of F-Actin filaments projected down from the dense F-Actin layer, extending about ~4 um depth into the cytoplasm, where smaller granules positive for Yl and Rab7 were located (Fig 3B). Further below this F-Actin web, where larger Rab7-positive and Yl-negative granules existed, small F-Actin puncta were scattered in the cytoplasm (arrows in Fig 3B). Interestingly, each Rab7-positive granule was always decorated by one or more of these F-Actin puncta that were often on the side of the granules proximal to the plasma membrane, a pattern that was especially apparent on smaller Rab7 granules (arrows in Fig 3B, see also Fig 3H and arrows in 3I, illustrated in 3J).

During endocytosis, a central regulator of actin dynamics is PI(4,5)P2, which is essential for the initiation of endocytosis, including the recruitment of AP-2 adaptor complex and Rab5 GTPase as well as other effectors that lead to the formation of CCPs and CCVs [20,21]. Similar as reported before [49], when detected by an ectopically expressed PLCδ1(PH)-GFP, a PI(4,5)P2 reporter composed of the Pleckstrin homology (PH) domain of the phospholipase-Cδ fused to GFP [64,65], it revealed a strong enrichment of PI(4,5)P2 on the plasma membrane of stage 10 oocytes, in a pattern similar to that of F-Actin, although the two did not completely overlap (Fig 3C, 3D, illustrated in 3E). Notably, even at relatively low expression levels, small patches of ectopically expressed PLCδ1(PH)-GFP could be found inside the oocytes that were always accompanied by aberrant F-Actin aggregation, a phenotype that was not observed in wildtype controls (arrows in Fig 3D1–3D2). The PLCδ1(PH)-GFP reporter can exert a dominant-negative effect by sequestering PI(4,5)P2 from its endogenous binding partners [66–69]. Indeed, at higher expression levels, larger PLCδ1(PH)-GFP-positive aggregates of different sizes accumulated together with F-Actin and Rab7 and strikingly long F-Actin filaments inside the oocytes (Fig 3F and 3G). Notably, similar phenotypes were observed in yeast defective for inositol(5)phosphatase *synaptojanin*, which controls the metabolism of PI(4,5)P2 [70]. Therefore, elevated levels of PLCδ1(PH)-GFP reporter sequester endogenous PI(4,5)P2 and blocks its efficient conversion, resulting in an abnormal buildup of PI(4,5)P2-associated actin regulators and stalled endosomal intermediates, including Rab7-positive late endosomes, inside the oocyte (Fig 3F and 3G). Together, these phenotypes support the potent role of PI(4,5)P2 on actin cytoskeleton dynamics and the importance of efficient phosphoinositide conversion in endosomal trafficking [20,21].

In contrast to the strong effect associated with ectopically expressed PI(4,5)P2 reporter, no apparent defects were observed in oocytes expressing 2xFYVE-GFP, the reporter for the early endosome marker PI(3)P [71,72]. Similar as Rab7, 2xFYVE-GFP was observed as numerous small puncta along the F-Actin-rich plasma membrane and also on the surfaces of larger granules near the cortex, but was absent on mature granules located deeper inside the oocyte (Figs 3H, 3I and S2A–S2C). Further, almost all these 2xFYVE-GFP-positive vesicles were also

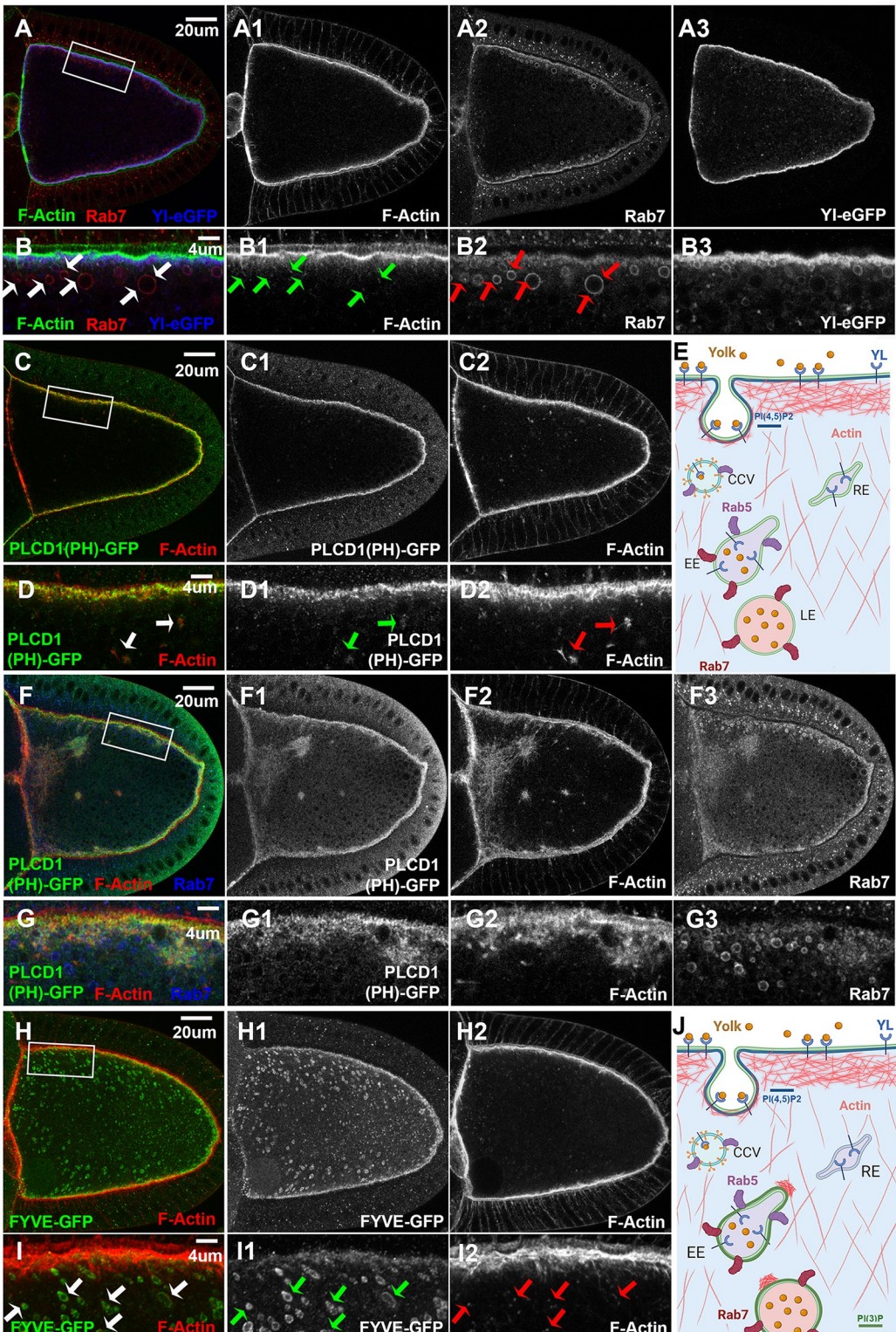

**Fig 3. Actin cytoskeleton dynamics and phosphatidylinositol phospholipids in vitellogenic oocytes.** Confocal microscopy of stage 10 egg chambers co-labeled with phalloidin for F-Actin and anti-GFP antibody for (A, B) Yl-eGFP-3xHA (blue), (C, D, F, G) PI(4,5)P2 reporter PLCδ1(PH)-GFP (green) and (H,I) PI(3)P reporter 2xFYVE-GFP (green), respectively, (A, B, F, G) together with endogenous Rab7, from flies with the corresponding reporter lines, as indicated. (A, C, F, H) Overview of the distribution of (A) Yl-GFP-3xHA, (C, F) PLCδ1(PH)-GFP and (H) 2xFYVE-GFP,

respectively, in the oocytes, as indicated. (B, D, G, I) High-magnification view of the corresponding cortex regions highlighted in (A, C, F, H), respectively, as annotated. (E) Cartoon illustration of the results from Fig 2 and Fig 3(A-D), showing the spatial relationship between Yl, PI(4,5)P2, F-Actin, Rab5 and Rab7 during endocytosis. (J) Cartoon illustration of the results from Fig 3, highlighting the association of F-Actin puncta with FYVE- and Rab7-positive granules. Genotypes: (A, B) $w^{1118}$; p{mini-W+, *yl*-eGFP-3xHA}. Note that the oocyte in Fig 3(A,B) is the same oocyte imaged in Fig 2A. Other images were from females flies heterozygous for both matalpha4-GAL-VP16 driver (BDSC #7062) and the following UAS-transgenic lines: (C, D) UASp-PLCδ1(PH).Cerulean.6 (BDSC #31421). (F, G) UASp-PLCδ11(PH).Cerulean.2 (BDSC #30895); (H, I) UAS-GFP-myc-2xFYVE (BDSC #42712); The sizes of the scale bars as labeled.

positive for Rab7 (S2A–S2C Fig), and comparable to Rab7-positive granules, were always decorated by one or more small F-Actin puncta that were located on the side proximal to the plasma membrane (arrows in Fig 3I, illustrated in 3J).

## Rab5 and Rab11 but not Rab4 and Rab7 are essential for yolk granule biogenesis

Using the established transgenic lines for normal and mutated forms of Rab proteins [73], we next examined the functional requirement of main Rab GTPases in yolk granules biogenesis. Consistent with the essential role of Rab5 in this process [49], oocytes overexpressing a YFP-tagged dominant negative (DN) Rab5 (YFP-Rab5-DN) contained only small vesicles visible under DIC imaging, with occasional presence of small Rab7-positive granules below the cortex (S3E and S3F Fig). Further, only granules at the very posterior, where the endocytosis is most active [46], were lysotracker-positive (S3G and S3H Fig, compared to wildtype S3C, S3D). The phenotypes induced by YFP-Rab5-DN were notably weaker than those reported for *rab5*-null flies [49], for example both F-Actin and Rab7-positive puncta were still enriched along the cortex. This is likely due to the presence of endogenous Rab5 that remains active in the oocytes over-expressing YFP-Rab5-DN. Nevertheless, they support the critical role of Rab5 in both endosomal fusion and maturation during granule formation. Interestingly, although the ectopically expressed YFP-Rab5-DN was diffusive throughout the cytoplasm, it still showed a relatively mild enrichment along the cortex (S3E1 and S3F1 Fig), indicating a local recruitment mechanism for Rab5 independent of its activity. Opposite of the phenotypes induced by YFP-Rab5-DN, constitutive active (CA) YFP-Rab5-CA induced large and abnormally shaped granules in the middle of the oocytes (S3I–S3L Fig), as reported in multiple previous studies [25,74]. These large granules were hybrid of early and late endosomes, as they were positive for lysotracker (S3K and S3L Fig compared to wildtype control in S3C, S3D) and their surface were decorated by both YFP-Rab5-CA, Rab7 and associated small F-Actin puncta (S3J1–S3J3-Fig). Notably, YFP-Rab5-CA and Rab7 also accumulated on many smaller aggregates inside the oocytes, but their levels along the cortex were significantly reduced (S3I and S3J Fig). In contrast, the pattern of F-Actin was not apparently affected, still showing normal enrichment along the cortex. As a control, oocytes over-expressing wildtype YFP-Rab5 showed no apparent defect, with both YFP-Rab5 and endogenous Rab7 enriched strongly along the cortex (S3M–S3N Fig). Together, they support that proper regulation of Rab5 is essential for granule biogenesis.

Oocytes expressing YFP-Rab11-DN showed severely disrupted vitellogenesis, as no recognizable yolk granules and hardly any recognizable Rab7-positive granules were present in these oocytes, which also appeared thinner at their posterior end, supporting a critical role of Rab11 in yolk granule biogenesis (Fig 4A and 4B). YFP-Rab11-DN was diffusively distributed in the cytoplasm, but still with a relatively stronger enrichment along the cortex (Fig 4B1). In addition, there was a reduced enrichment of Rab7 along the cortex, which also became more diffusive at the posterior, no longer anchored along the cortex (Fig 4A2, compare with Fig 2A).

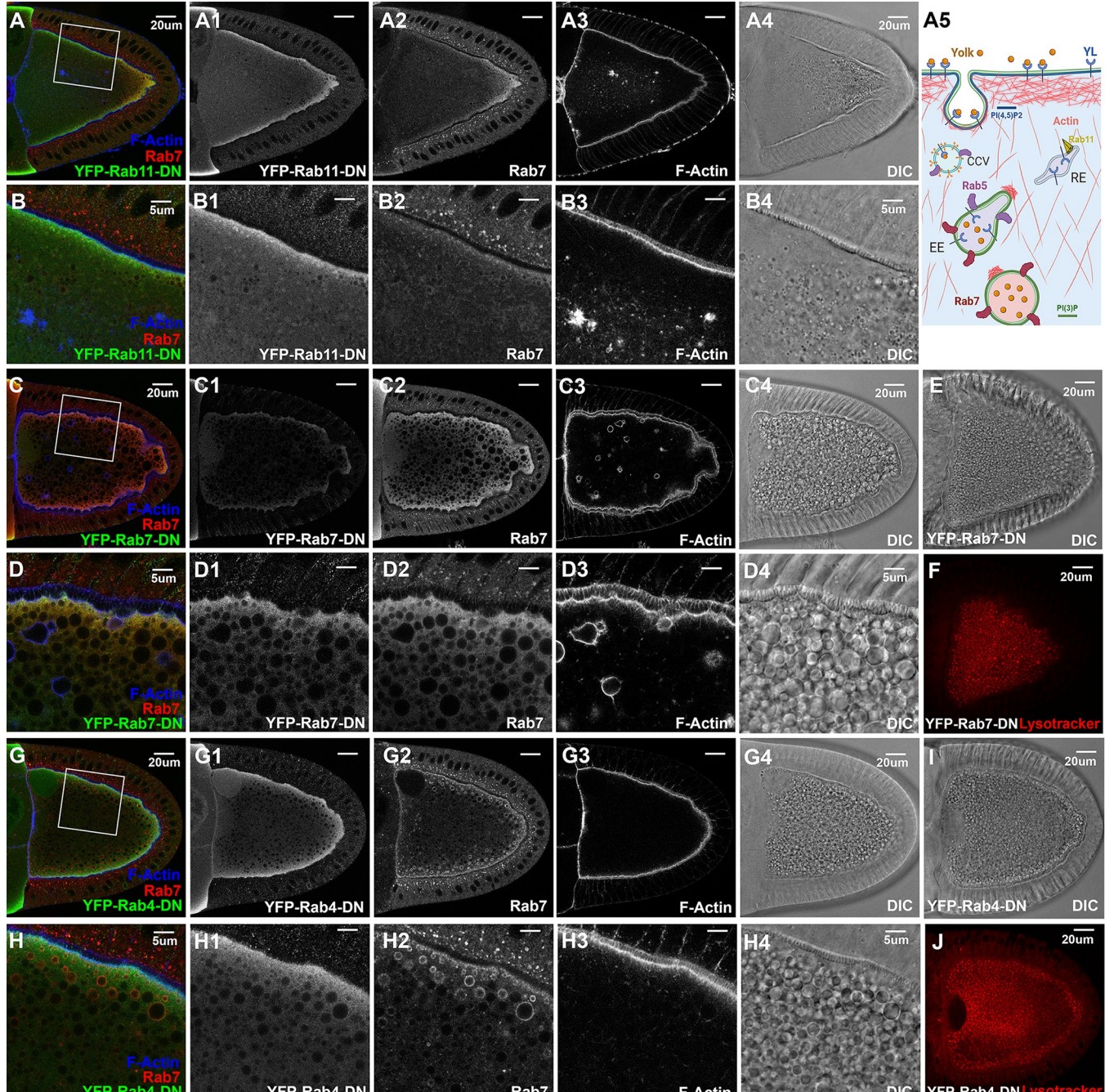

**Fig 4. Rab4, Rab5, Rab7 and Rab11 in yolk granule biogenesis.** Confocal fluorescent microscopy and DIC imaging of stage 10 egg chambers with oocyte-specific expression of dominant negative (A-B4) YFP-Rab11-DN, (C-F) YFP-Rab7-DN and (G-J) YFP-Rab4-DN, respectively, that were co-stained by anti-GFP antibody (green), anti-Rab7 (red) and phalloidin for F-Actin (blue), presented as overlaying images in color or as individual channels in gray, or (E, F, I, J) by lysotracker stain alone (red), as annotated. (B-B4, D-D4, H-H4) High-magnification views of the cortex regions highlighted in (A, C, G), respectively, as indicated. Genotypes: The samples were from adult females flies heterozygous for both matalpha4-GAL-VP16 driver (BDSC #7062) and the following UAS-transgenic lines: (A, B) Rab11-DN: UASp-YFP.Rab11.S25N (BDSC #9792). (C-F). Rab7-DN: UASp-YFP.Rab7.T22N (BDSC #9778). (G-J). Rab4-DN: UASp-YFP.Rab4.S22N (#9768). The sizes of the scale bars as annotated inside images.

Lastly, although F-Actin showed relatively normal enrichment along the cortex, aberrant F-Actin aggregates were present inside the oocytes. But unlike the abnormal endosomal structures induced by ectopic PLCδ1(PH)-GFP(Fig 3F and 3G), these F-Actin aggregates were negative for Rab7 and not associated with long actin filaments. In contrast to overexpressed YFP-Rab11-DN, neither constitutive active nor wildtype (WT) YFP-Rab11 disrupted granule biogenesis when overexpressed (S4 Fig). The ectopically expressed WT YFP-Rab11 was sharply concentrated in small puncta along the cortex, in contrast to the more diffusive presence of YFP-Rab11-CA. Neither YFP-Rab11-CA nor WT YFP-Rab11 showed apparent association with Rab7-positive granules.

Surprisingly, despite Rab7's proposed role as a major regulator of endosomal growth and maturation, vitellogenesis did not appear to be significantly affected in stage 10 oocytes overexpressing either DN-, WT- or CA-YFP-Rab7, as large numbers of lysotracker-positive yolk granules were present in these oocytes (Figs 4C–4F and S5), echoing the results of a recent study [74]. As expected, YFP-Rab7-WT was enriched near the cortex region and notably decorated the surface of some small- and large-sized lysotracker-positive granules (S5C and S5D Fig). Similarly, YFP-Rab7-CA was enriched as puncta along the cortex and on the surface of larger granules nearby (S5E and S5F Fig), while YFP-Rab7-DN was diffusive in the cytoplasm and not located on the surface of the granules (Fig 4D). Unexpectedly, in oocytes expressing YFP-Rab7-DN, some large granules were shrouded in an F-Actin layer, with some granules in irregular shape while others had F-Actin extensions projecting from the granule surface (Fig 4D3), indicating a role for Rab7 in actin regulation during endosomal maturation.

Lastly, Rab4 is well-documented for its role as a major regulator of fast recycling in endosomal trafficking [75]. However, perturbation of Rab4 in oocytes by overexpressing either DN-, CA- or WT- Rab4 did not appear to affect vitellogenesis, as F-Actin and Rab7 distribution, granule formation and acidification all proceeded normally (Figs 4G–4J and S6). These observations are in line with the relatively low levels of endogenous Rab4 inside the oocyte and its lack of enrichment at the cortex (Fig 2D), and consistent with the recent findings that *rab4* knockout flies are homozygous viable and fertile, in contrast to the essential roles of Rab5, Rab7 and Rab11 in fly development [76], supporting its dispensable role in vitellogenesis. Surprisingly, while both the ectopically expressed DN- and CA-YFP-Rab4 were distributed diffusively throughout the cytoplasm, ectopic YFP-Rab4-WT overlapped significantly with endogenous Rab7 (Figs 4G–4H and S6), reminiscent of the association of endogenous YFP-Rab4 with Rab7-positive endosomes (Fig 2D).

Together, these results, summarized in Table 1, support the critical roles of Rab5 and Rab11 in yolk granule biogenesis, and imply a potential role for Rab7 in regulating actin dynamics.

## An *in vivo* RNAi screen to evaluate Rab5 effectors in yolk granule biogenesis

The above results establish the vitellogenic stage oocyte as a potential *in vivo* model for studying endocytosis and endolysosomal trafficking. Taking advantage of the existing collections of genome-wide transgenic RNAi lines [77–79], we analyzed a selected group of Rab5 effectors identified in a recent proteomic study [58] for their roles in yolk granule biogenesis. To bypass the animal lethality associated with global knockdown of essential genes, we chose matalpha4-GAL-VP16, a maternal-specific GAL4 line that expresses a strong transcriptional activator GAL4-VP16 fusion only in germline oocyte and nurse cells under the control of alphaTub67C promoter [80]. Since this matalpha4-GAL line initiates dsRNA expression around stage 5 of egg chamber development [74], it should avoid potential disruption on earlier oocyte

**Table 1.  Summary of phenotypes in oocytes overexpressing YFP-tagged Rab4, Rab5, Rab7 and Rab11 proteins.**

| UASp-YFP-Rab Lines | BDSC stock # | Phenotypes of stage 10 oocytes overexpressing YFP-Rab proteins driven by matalpha4-GAL-VP16 driver | | | | |
|---|---|---|---|---|---|---|
| | | Subcellular localization of YFP-Rab protein | Rab7 | Granule Size | F-Actin | Lysotracker |
| Rab4-WT | 9767 | Co-localize with Rab7-positive granules, also diffusive in cytoplasm | Normal | Normal | Normal | Normal |
| Rab4-CA | 23268 | Diffusive, with enrichment near oocyte cortex | Normal | Normal | Normal | Normal |
| Rab4-DN | 9768 | Diffusive, with enrichment near oocyte cortex | Normal | Normal | Normal | Normal |
| Rab5-WT | 24616 | Strongly enriched as puncta at cortex | Normal | Normal | Normal | Normal |
| Rab5-CA | 9774 | Enriched in puncta and on the membrane surface of large granules. Reduced enrichment at oocyte cortex. | Enriched in puncta and on the membrane surface of large granules. Reduced enrichment at oocyte cortex. | Very large | Normal near the cortex, more smaller F-actin puncta associated with large YFP-Rab5-CA granules inside the oocytes | Large granules are lysotracker-positive. |
| Rab5-DN | 9778 | Diffusive, with enrichment near oocyte cortex | Enrichment at oocyte cortex. More diffusive Rab7 in the cytoplasm. Few large Rab7-positive granules exist | Small granules | Normal | Existence of few very small lysotracker-positive granules near the posterior of oocytes |
| Rab7-WT | 23641 | Enriched at oocyte cortex and on the surface of Rab7-positive granules | Normal | Normal | Normal | Normal |
| Rab7-CA | 24103 | Enriched at oocyte cortex and on the surface of Rab7-positive granules | Normal | Normal | Normal | Normal |
| Rab7-DN | 9778 | Diffusive, with enrichment near cortex | Diffusive | Large | Aberrant "rings" on some Rab7-positive granules | Normal |
| Rab11-WT | 50782 | Enriched as small puncta at oocyte cortex. | Normal | Normal | Normal | Normal |
| Rab11-CA | 23260 | Diffusive, with enrichment near oocyte cortex | Normal | Normal | Normal | Normal |
| Rab11-DN | 9792 | Diffusive, with enrichment near oocyte cortex | Diffusive, reduced enrichment at oocyte cortex, increased levels near the posterior of the oocytes | Few visible and very small-sized granules | Aberrant small F-actin aggregates inside the oocytes | Existence of few small lysotracker-positive granules near the posterior of oocytes |

development or complication of animal lethality while might still afford sufficient time for RNAi-mediated knockdown of target genes by stage 8, when vitellogenesis starts.

Among the 36 RNAi lines we tested that targeted 22 unique reported Rab5 effectors (Table S1) [58], only a few caused distinct phenotypes. Among them, oocytes expressing dsRNA against PI3K59F contained hardly any visible granules (Figs 5A4 and 5B4), and lysotracker staining revealed the presence of only very few and small-sized lysotracker-positive structures, which were also surrounded by weak Yl-GFP signal, similar as that observed in wildtype control (compared Supplemental S7D–S7F Fig to control S7A–S7C). PI3K59F encodes the fly homolog of VPS34, the class III PI3-kinase that is required for the endosomal production of PI(3)P [24]. Consistently, these oocytes contained few small FYVE-GFP-positive granules, which only partially co-localized with, but not completely overlapped, with Rab7 (Figs 5C, 5D and S2D–S2F, compare to WT controls in Figs 3H, 3I and S2A–S2C). Moreover, Yl-GFP was no longer concentrated on the plasma membrane but became dispersed in membranous structures throughout the oocyte (Figs 5A, 5B, S7D–S7F, S7I and S7J). Rab7 was similarly disrupted, showing no enrichment along the plasma membrane and no visible

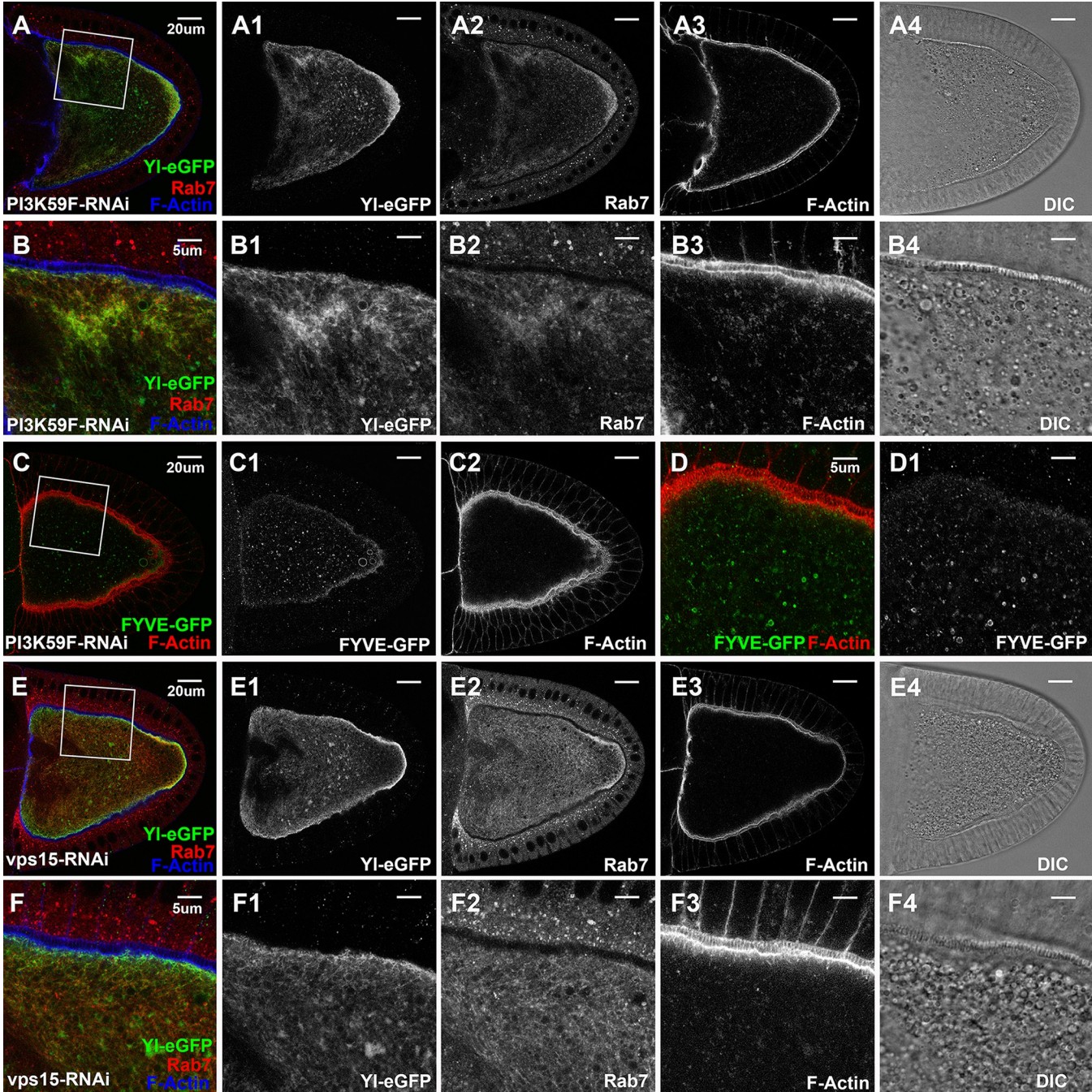

**Fig 5. Essential roles of VPS34/VPS15 PI3 kinase complex in Yl recycling and yolk granule biogenesis.** Confocal fluorescent microscopy and DIC imaging of stage 10 egg chambers with oocyte-specific expression of dsRNA again (A-D1) VPS34/PI3K59F or (E, F) Vps15, from flies (A-B4, E-F4) carrying genome-tagging Yl-eGFP-3xHA, co-labeled with antibodies against GFP (green), endogenous Rab7 (red) and phalloidin for F-Actin (blue), or (C-D1) expressing 2xFYVE-GFP, co-labeled for F-Actin (red), presented as overlaying images in color or as individual channels in gray, as annotated. (B, D, F) High-magnification view of the cortex regions highlighted in (A, C, E), respectively, as indicated. Genotypes: The samples were from adult females flies heterozygous for both matalpha4-GAL-VP16 driver (BDSC #7062) and the following UAS-transgenic RNAi and Yl- or FYVE-reporter lines: (A,B) P{TRiP.HMJ30324}attP40 (#64011); p{mini-W+, *yl*-eGFP-3xHA}. (C,D) P{w(+mC) = UAS-GFP-myc-2xFYVE}2 (#42172); P{TRiP.HMS00261}attP2 (#33384); (E,F). P{TRiP.GL00085} attP2 (#35209)/ p{mini-W+, *yl*-eGFP-3xHA}. The sizes of the scales as annotated inside images.

Rab7-positive granules, but became similarly sequestered with Yl inside the oocyte, including an abnormal enrichment near the posterior (Fig 5A, 5B, and Supplemental S7I, S7J). Further, in regions with ectopic enrichment of Yl-GFP and Rab7, there was a similar increased accumulation of small F-Actin puncta, although F-Actin enrichment on plasma membrane was largely normal (Fig 5B1–5B3). Closer inspection revealed that the internally sequestered Yl-GFP did not overlap with the remaining lysotracker-positive structures (S7D–S7F Fig), and although Yl and Rab7 showed a similar pattern of sequestration inside the oocyte, they only partially overlapped (S7J Fig).

Importantly, PI3K59F forms an active class III PI3K complex by dimerizing with its obligate partner VPS15 [81,82], and oocytes with RNAi-mediated depletion of VPS15 manifested almost identical phenotypes as those seen in PI3K59F knockdown, including a lack of yolk granules and the abnormal accumulations of Yl-GFP, Rab7 and F-Actin inside the oocytes (Fig 5E and 5F). Given that cortical F-Actin was largely normal in PI3K59F- and VPS15-knockdown oocytes (Fig 5A3, 5B3 compared to 5E3, 5F3), similar as oocytes expressing YFP-Rab5-DN (S3E and S3F Fig), they are consistent with the model that the VPS34/VPS15 kinase complex functions downstream of PI(4,5)P2 conversion and is critical for endosomal PI(3)P production, which is essential for the subsequent endosomal fusion, sorting and recycling. Diminished levels of PI(3)P compromises the sorting and trafficking in early endosomes, resulting in blocked Yl recycling and its abnormal accumulation along with F-Actin and Rab7 within the stalled endosomal structures.

CCZ1 is a Rab5 effector critical for the recruitment of Rab7 onto Rab5-positive endosomes [27,28]. Consistently, in oocytes expressing dsRNA against CCZ1, Rab7 showed significantly reduced enrichment along the cortex and most strikingly, was no longer associated with the granule membrane but became diffusive in the cytoplasm (Fig 6, compared with Rab7 in wild-type Figs 2A, S2A–S2C, and S3A, S3B). However, CCZ1-depleted oocytes showed a normal distribution of both F-Actin (Fig 6A2, 6B2) and Yl-GFP (Fig 6A3, 6B3), with normal presence of PI(3)P-positive granules (Fig 6C and 6D), and numerous granules (Fig 6A4 and 6B4). These phenotypes are consistent with the role of CCZ1 in the endosomal recruitment of Rab7 and

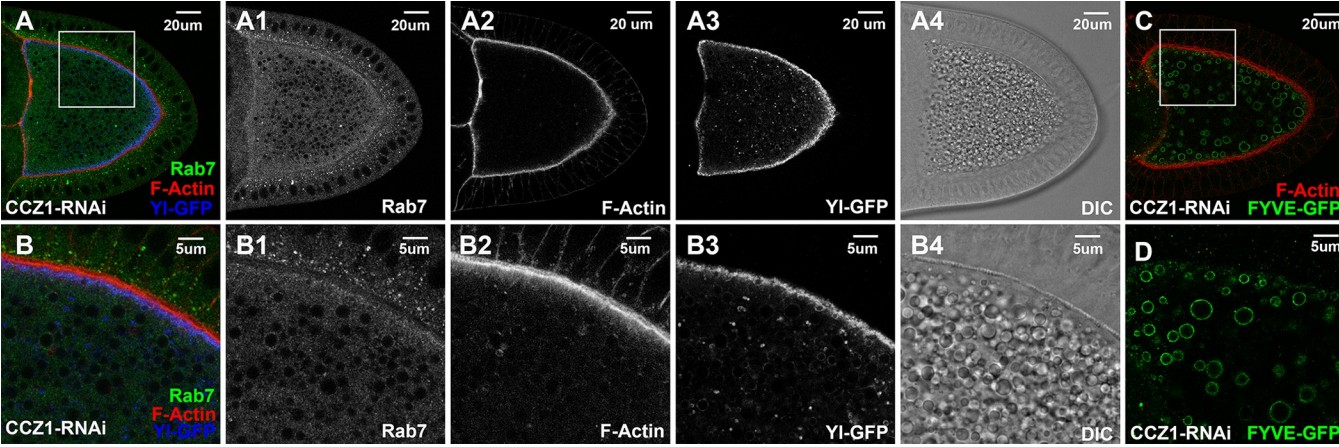

**Fig 6. The roles of CCZ1 in Yl recycling and yolk granule biogenesis.** Confocal fluorescent microscopy and DIC imaging of stage 10 egg chambers with oocyte-specific expression of dsRNA against CCZ1 from flies (A-B4) carrying genome-tagging Yl-eGFP-3xHA reporter, co-labeled with phalloidin (red) and antibodies against endogenous Rab7 (green) and GFP for Yl (blue), or (C, D) with ectopic expression of 2xFYVE-GFP, co-labeled with antibody against GFP (green) and phalloidin (red), presented as overlaying images in color or as individual channels in gray, as annotated. (B-B4) and (D) are high-magnification views of the cortex regions highlighted in (A) and (C), respectively. Genotypes: The samples were from adult females flies heterozygous for both matalpha4-GAL-VP16 (BDSC #7062) and the following PI(3)P or Yl and UAS-transgenic RNAi lines: (A,B) P{TRiP.HMJ24129}attP40 (#62889); p{mini-W+, *yl*-eGFP-3xHA}. (C,D) P{w(+mC) = UAS-GFP-myc-2xFYVE}2 (#42172), P{TRiP.HMJ24129}attP40 (#62889). The sizes of the scale bars as annotated.

interestingly, also with the observation that perturbation of Rab7 did not significantly disrupt granule biogenesis (Fig 4C–4F).

## RAVE complex and V-ATPase vacuolar proton pump control the early endosome formation

Rabconnectin-3A (Rbcn-3A) is another Rab5 effector isolated from the proteomic study [58]. In oocytes expressing dsRNA against Rbcn-3A, there were few granules visible under DIC (Fig 7A3, 7B3) and few Rab7-positive granules detectable under confocal microscope. Instead, an abnormally high level of Rab7 became accumulated near the posterior region of the egg chamber (Fig 7A). Similarly, there were few large FYVE-GFP-positive granules (Fig 7C compared to control Figs 3H and S2A–S2C). Instead, FYVE-GFP accumulated as small puncta

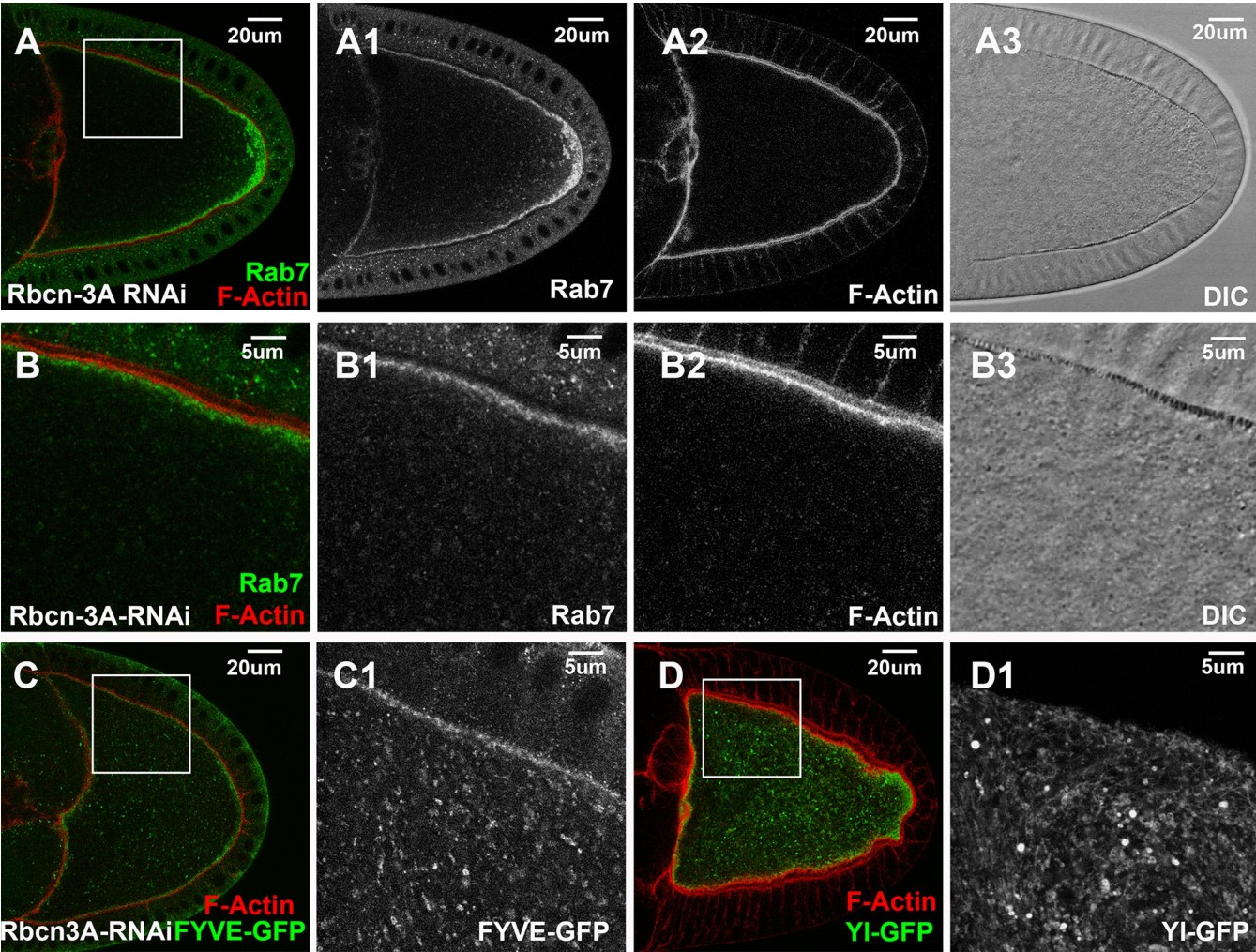

**Fig 7. Rbcn-3A is essential for Yl recycling and yolk granule biogenesis.** Confocal fluorescent microscopy and DIC imaging of stage 10 egg chambers with oocyte-specific expression of dsRNA against Rbcn-3A, double-labeled with phalloidin (red) and antibody against (A, B) endogenous Rab7 (green) or (C, D) GFP (green) from flies (C) with ectopic expression of GFP-2xFYVE or (D) carrying a genome-tagging Yl-eGFP-3xHA. Images are presented as overlaying images in color or in individual channels in gray, as annotated. (B-B3, C1, D1) High-magnification views of the cortex regions highlighted in (A, C, D), respectively, as annotated. Genotypes: The samples were from adult females flies heterozygous for both matalpha4-GAL-VP16 driver (BDSC #7062) and the following UAS-transgenic RNAi and PI(3)P or Yl reporter lines: (A,B) P{TRiP.HMS01287}attP2 (#34612). (C) P{w(+mC) = UAS-GFP-myc-2xFYVE}2 (#42172); P{TRiP. HMS01287}attP2 (#34612). (D). P{TRiP.HMS01287}attP2 (#34612); p{mini-W+, *yl*-eGFP-3xHA}. The sizes of the scale bars as annotated.

inside the cytoplasm (Fig 7C1). However, F-Actin, Rab7 and FYVE-GFP still showed enrichment along the cortex. Further examinations revealed a severe disruption of Yl distribution, as Yl was no longer enriched in puncta along the cortex but was trapped in membranous structures inside the oocyte (Fig 7D). The observed phenotypes in Rbcn-3A knockdown bore some similarity with those observed in oocytes depleted of VPS34/VPS15 PI3K kinase complex (Fig 5), including the absence of recognizable yolk granules and trapped Yl receptor, although Rab7 and F-Actin were not as severely affected, and 2xFYVE-GFP signals were still present but accumulated in small puncta (Fig 7 compared to Figs 5 and S2D–S2F).

Rbcn-3A is a component of the evolutionarily conserved RAVE (Regulator of H+-ATPase of Vacuolar and Endosomal membranes) complex, which functions to assist the proper assembly of integral membrane V0 multi-subunit subcomplex and cytosolic V1 ATPase subcomplexes into an active vacuolar proton-translocating ATPase (V-ATPase) holo-enzyme on the vesicular membrane [83]. As V-ATPase is required for the proton uptake into the lumens of the residing organelles and their acidification [84,85], the strong effects of Rbcn-3A depletion would predict a similar functional requirement of the V1 and V0 ATPase subcomplexes in yolk granule biogenesis. To test this, we examined RNAi lines targeting eight different subunits in V0 and V1 subcomplexes (Vha68-2, Vha36-3, Vha26, Vha55, Vha100-1, Vha100-2, VhaAC45RP, VhaAC39-2). Among them, oocytes expressing dsRNA against Vha26 (Fig 8A and 8B), an E subunit of the V1 subcomplex, or Vha68-2 (Fig 8C–8F), subunit 2 of the V1 subcomplex, exhibited phenotypes similar to those induced by Rbcn-3A knockdown, including the absence of yolk granules, trapped Yl receptor and loss of Rab7- and FYVE-positive endosomes inside the oocytes (Fig 8). Together, these findings support that the recruitment of RAVE complex and the subsequent assembly of active V-ATPase complex are essential early steps in endosomal trafficking events that lead to the biogenesis of mature yolk granules.

## Discussion

Our results support that yolk granule biogenesis largely follows the canonical endolysosomal trafficking processes, regulated through cascades of effectors that coordinate and execute different steps of endosomal dynamics, including Rab5 and Rab11 in endocytosis and recycling, the VPS34/VPS15 PI3K kinase complex and PI(3)P production in endosomal biogenesis, and the Rab5 to Rab7 conversion regulated through the CCZ1/Mon1 complex. Together, they demonstrate that the vitellogenic oocyte is an excellent *in vivo* model for studying receptor-mediated endocytosis and subsequent endolysosomal trafficking in a multicellular system under physiological conditions.

### Roles of Rab4 and Rab7 in endolysosomal trafficking

One surprising finding from the study is the dispensable roles of Rab4 and Rab7 in yolk granule biogenesis. It is generally accepted, mainly from studies in mammalian cells, that Rab4 mediates the fast recycling directly from early endosomes back to the cell surface, while Rab11 controls the slow recycling that often transits through peri-nuclear endocytic compartment to the plasma membrane [14–19]. Given the heavy intensity of Yl recycling that happen primarily near the plasma membrane during vitellogenesis (Fig 1), it is unexpected that disturbances of Rab4 had no apparent effect on Yl recycling and yolk granule biogenesis (Figs 4G–4J and S6). More intriguingly, YFP-Rab4 is distributed diffusively as small puncta and co-localized with Rab7-positive granules (Fig 2D). Importantly, such localization patterns are unlikely to be artifacts, as endogenous Rab4 in surrounding follicle cells also clearly co-localized with Rab7 on the same endosomal structures (Fig 2D). Lastly, ectopically expressed wildtype YFP-Rab4 prominently localized on Rab7-positive endosomes (S6B Fig). These observations are rather

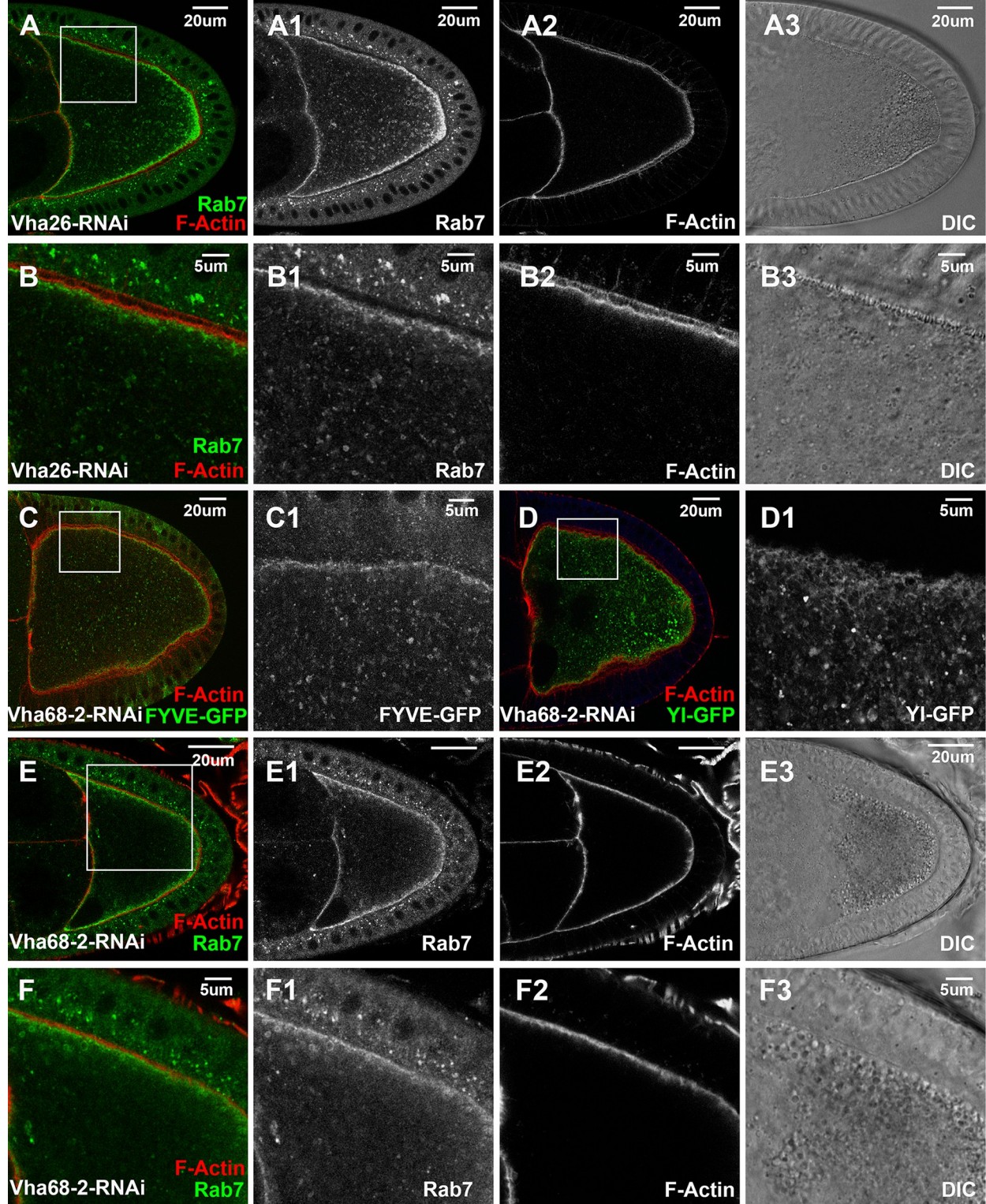

**Fig 8. V-ATPase is essential for Yl recycling and yolk granule biogenesis.** Confocal fluorescent microscopy and DIC imaging of stage 10 egg chambers with oocyte-specific expression of dsRNA against (A, B) Vha26a or (C-F) Vha68-2 double-labeled for F-Actin (red) and (A, B, E, F) endogenous Rab7 (green) or (C, D) GFP (green) from flies (C) with ectopic expression of 2x-FYVE-GFP or (D) carrying genome-tagging Yl-eGFP-3xHA reporter. (B, C1, D1, F) High-magnification views of the cortex regions highlighted in (A, C, D, E), respectively, as annotated. Genotypes: The samples were from adult females flies heterozygous for both matalpha4-GAL-VP16 driver (BDSC #7062) and the following UAS-

transgenic RNAi and PI(3)P or Yl reporter lines: (A, B) P{TRiP.HMS01912}attP2 (#38996). (C) P{w(+mC) = UAS-GFP-myc-2xFYVE}2 (#42172); P{TRiP.HMS01056}attP2 (#34582). (D). P{TRiP.HMS01056}attP2 (#34582);p{mini-W+, *yl*-eGFP-3xHA}. (E, F) P{TRiP.HMS01056}attP2 (#34582). The sizes of the scale bars as annotated.

surprising, as in mammalian cells, Rab4 and Rab7 were shown to locate on non-overlapping endosomal compartments and with different endosomal functions [86]. However, a later study reported that Rab4 was involved in both recycling and degradative endosomal trafficking [87]. Altogether, they raise an intriguing possibility that Rab4 and Rab7 overlap spatially and functionally during endolysosomal trafficking and that similar mechanisms control their endosomal recruitment. More studies are needed to determine whether the Rab4/Rab7 colocalization is cell type-specific and exclusive to *Drosophila*, and to clarify their spatial and functional relationships.

Similarly, Rab7 has been accepted as a key regulator of endosomal maturation, controlling various aspects of endosomal dynamics, such as fusion among late endosomes and with lysosomes, cargo retrieval by retromer and transport by microtubule-based motors [4,14,88,89]. However, manipulations of Rab7 did not significantly affect yolk granule biogenesis, as vitellogenic oocytes were filled with acidified granules of relatively normal sizes (Figs 4C–4F and S5). Consistently, in CCZ1-knockdown oocytes, which blocked the recruitment of Rab7 to the endosomal membrane, yolk granule biogenesis also appeared to proceed normally (Fig 6). Together, they raise a tantalizing possibility that, at least under this physiological setting, Rab7 might be dispensable for endosomal maturation. Notably, in an earlier study in HeLa cells, Rab7 depletion similarly did not affect late endosome biogenesis but blocked the fusion of late endosome with the lysosome [90]. Therefore, one possible scenario is that during yolk granule biogenesis, although Rab7 is recruited early onto endosomal membrane, it is primarily required at much later stages, after the growth and maturation of late endosomes, to promote their fusion with lysosomes. In this scenario, it is important to note that, although yolk granules are assumed to be specialized latent lysosomes containing inactive lysosomal enzymes [34–37], we still lack the molecular tools to definitively define whether the yolk granules we observed under confocal microscope are lysosomes or just late endosomes. Alternatively, the dispensable role of Rab7 in granule biogenesis could be due to redundant mechanisms involving Rab7 and other GTPases such as Arl8b and Rab2 as well as the HOPS complex, which together coordinate the endosomal maturation and their fusion with lysosomes [14].

Interestingly, in oocytes expressing dominant negative Rab7, large granules with a dense F-Actin coating were observed (Fig 4C and 4D), suggesting a role of Rab7 in preventing F-Actin accumulation on late endosomal membrane. In both yeast and mammalian cells, membrane assembly of actin filaments is required for the fusion between mature endosomes and lysosomes [91–94]. Further, in yeast, endosomal actin polymerization is catalyzed by Rho GTPase Rho1p and Cdc42, which in turn are activated by Ypt7p, the yeast Rab7 homologue [95]. It remains to be clarified whether Rab7 engages in similar conserved mechanisms to regulate actin dynamics.

## Dynamics of actin cytoskeleton in endocytosis

Actin dynamics are involved in various aspects of endocytosis and endolysosomal trafficking, including the internalization of endocytic vesicles, endosome motility and potentially the fusion between late endosomes and lysosomes [4,63,96]. Consistently, distinct actin structures that correlate with different stages of endosomal trafficking exist in the vitellogenic oocyte, including a dense F-Actin layer that overlaps significantly with Yl receptor on the plasma membrane, an adjacent thin web of F-Actin projecting down into the cortex, and scattered

small F-Actin puncta decorating the FYVE- and Rab7-positive granules that emerge from the cortex area (Fig 3). Notably, these small F-Actin puncta were mostly localized on the side of the granules proximal to the cortex (Fig 3B and 3I), reminiscent of the actin tails on motile endosomes in mammalian cells [97–101], suggesting their potential role in propelling endosomes into the interior of the oocytes, in a mechanism similar to the proposed "actin-rocketing" model that drives the motility of invading pathogens in infected cells [102].

The cortical F-Actin dynamics are likely regulated by phospholipid PI(4,5)P2, which has well-established roles in recruiting actin regulators such as WASP-family proteins and Arp2/3 complex [103–107], and is enriched on oocyte membrane where it overlaps with F-Actin (Fig 3C and 3D). Consistently, higher levels of ectopic PLCδ1(PH)-GFP reporter, which can exert a dominant negative effect by sequestering PI(4,5)P2 [66–69], severely disrupted endocytosis and caused the abnormal congregation of endosomal structures with Rab7 and long F-actin filaments (Fig 3F and 3G).

## RAVE and V-ATPase complexes are essential during early endosomal formation

The highly conserved V-ATPase complex controls endosomal acidification, which is essential for cargo sorting, membrane trafficking and endosomal maturation [4,108]. The reversible disassembly and regulated re-assembly of functional V-ATPase is controlled by RAVE complex in yeast and Rabconnectin-3 complexes in metazoan [83]. Consistently, oocyte-specific depletion of either Rbcn-3A (Fig 7) or subunits of V-ATPase complex (Fig 8) caused similar granule biogenesis defects, in particular a blocked Yl receptor recycling and endosome formation, in line with their critical role for efficient endocytosis and endosomal trafficking. Notably, Rbcn-3A was isolated as a Rab5 effector in a proteomic study from *Drosophila* S2 cells [58], suggesting that Rab5 recruits V-ATPase onto earlier endosomes via the RAVE complex, which in turn leads to the acidification of early endosomes and the subsequent release of Yolk cargo from Yl receptor. In the absence of functional RAVE and V-ATPase complexes, Yolk cargo can not be efficiently freed from Yl receptor in a PH-dependent manner, resulting in blocked Yl recycling and aborted yolk granule formation.

## A model for yolk granule biogenesis

In line with the current models of endocytosis and endosomal trafficking [1–5,14], our results support the following events that control yolk granule biogenesis in *Drosophila* oocytes (Fig 9). During vitellogenesis, Rab5 promotes the extensive rounds of clathrin-mediated endocytosis of Yolk proteins by Yl receptor, a process that is facilitated by dynamic actin cytoskeleton networks regulated by PI(4,5)P2-associated actin regulators. After the removal of clathrin and depletion of PI(4,5)P2 and associated factors from the endocytosed vesicles, membrane-associated Rab5 recruits a collection of effectors that set off a cascade of downstream events, including the VPS34/VPS15 PI3 kinase complex for local production of PI(3)P and RAVE/Rabconnectin-3 complex that facilitates the stable assembly of V-ATPase complex on early endosomes. The conversion of PI(4,5)P2 to PI(3)P on endosomal membrane is accompanied by the replacement of branched actin filaments with small actin nucleations on the distal side of granule surface to propel their inward movement. Simultaneously, Rab5 synergizes with PI(3)P on the early endosomal membranes to recruit PI(3)P-binding effectors such as Hrs and other ESCRT components and CCZ1/Mon1 complexes to catalyze endosomal fusion, sorting, Rab5 to Rab7 conversion and maturation, while acidification of endosomal lumen by V-ATPase leads to the separation of Yl from Yolk cargo. The ligand-free Yl is subsequently segregated away and enriched in the tubular membrane extension where it is quickly recycled

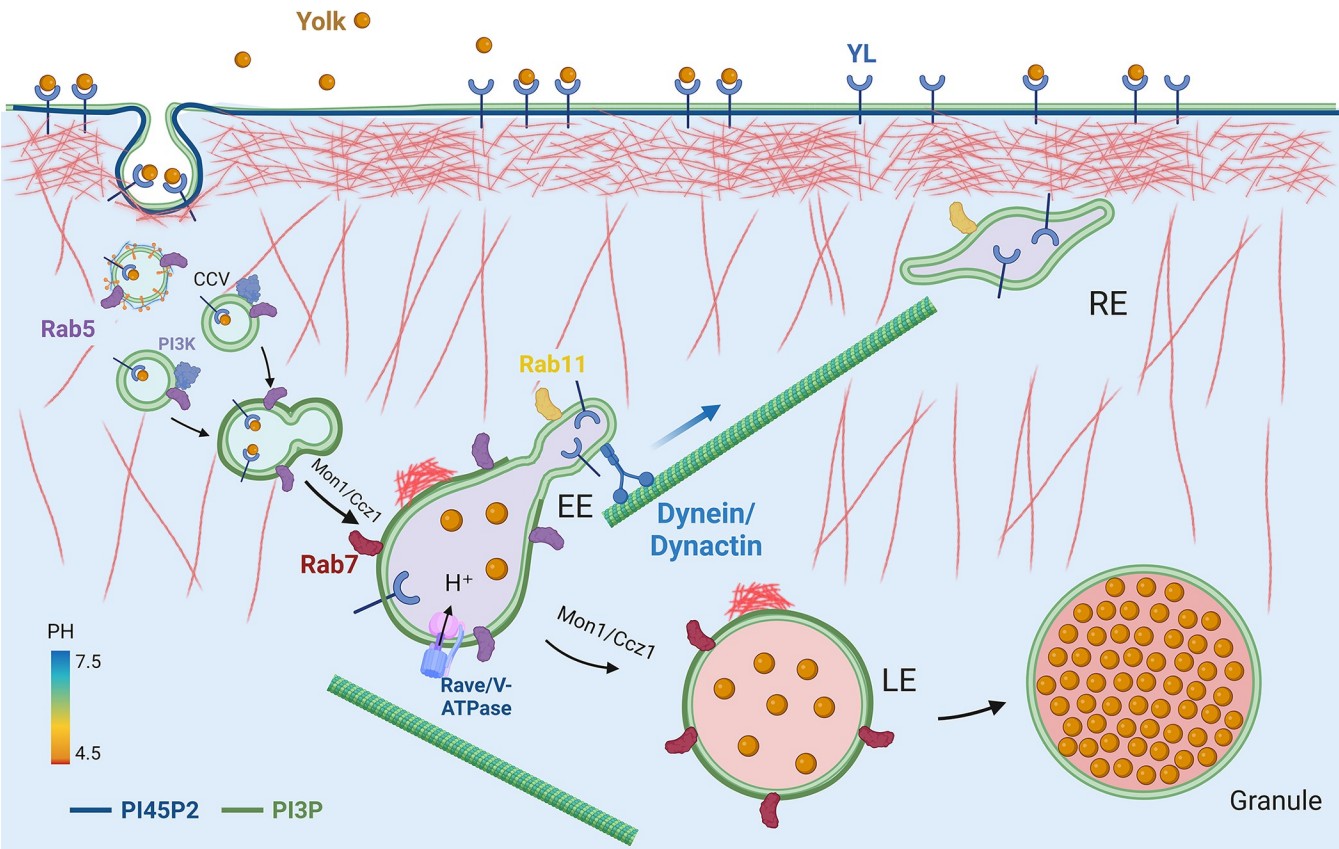

**Fig 9. Model of yolk granule biogenesis.** See text in the Discussion for details.

back to the plasma membrane by Rab11 for new rounds of endocytosis, while Yolk proteins are retained in Rab7-positive vacuolar domain. Through concerted action by Rab7 and other functionally redundant cytoskeleton and membrane regulators to coordinate actin dynamics with membrane fusion, Yolk-containing endosomes undergoes continuous fusion and maturation processes while being transported into the interior of the oocyte through actin- and/or microtubule-based mechanisms, and eventually fuse with the lysosome to complete their maturation.

Many questions arise from this study, such as the exact spatial and functional relationship between Rab4 and Rab7 in endolysosomal trafficking, and when and which GTPase-activating proteins (GAPs) control the release of Rab7 from mature granules [109,110], among others. Many factors would affect the interpretation of our observations, such as the limited resolution of subcellular structures under conventional confocal microscope. Notably, most of the genes tested in our RNAi screen did not show significant effect on granule biogenesis (Table S1), which could be due to inefficient depletion of the targeted protein products in stage 10 oocytes by RNAi, either because of long perdurance of the studied proteins or due to ineffective RNAi constructs, or both, especially considering the relatively short duration between stage 5 when matalpha4-GAL-VP16 becomes active to drive dsRNA expression and stage 10 when oocytes are analyzed. In support of this, using the same driver, we found that RNAi against Rab4 did not have apparent effect on granule biogenesis, as expected, RNAi against Rab5 induced robust phenotypes similar as or even more severe than that by Rab5-DN, but a RNAi line against Rab7 only partially knocked down endogenous Rab7 protein (S8 Fig). Together, they suggest

variable knockdown efficiency by different RNAi lines, which also implies that only a fraction of the RNAi lines tested in our pilot screen induced significant loss of function phenotypes. Nevertheless, several of them, including PI3K59/VPS15, Rbcn3A and Rab5, showed distinct granule biogenesis defects when driven by matalpha4-GAL-VP16 (Figs 5–8). When directed by ubiquitous tubulin-GAL4, we found that the same RNAi lines for these essential genes caused animal lethality, preventing their functional evaluation in the oocytes. Thus, although not 100% effective in depleting the target proteins in stage 10 oocytes, dsRNA expression driven by matalpha4-GAL-VP16 can still be a valuable option to quickly survey large number of essential genes for their roles in yolk granule biogenesis. Development of more potent and oocyte-specific GAL4 drivers will greatly facilitate similar studies in the future.

Our simplified model will require further validation and revision at molecular and subcellular levels by incorporating new imaging tools such as live- and super-resolution microscopy and by employing additional markers specific for different endosomal trafficking steps, such as lysosomal markers that can distinguish between maturating and matured granules. Moreover, as our studies primarily focused on stage 10 oocytes, it will be important to test to what extent the findings from this unique time point can be applied to membrane trafficking in *Drosophila* oocytes at other developmental stages, to other cell types and tissues such as neurons, and especially to other organisms such as mammals that have more redundant genes (e.g., three Rab5-like genes in human genome as comparted to a single Rab5 gene in the fly) [76] and likely more complicated regulatory circuitries on endosomal trafficking to meet their more complex physiological demands.

## Materials and methods

### *Drosophila* husbandry and genetics

Fly stocks were maintained at room temperature following standard culture conditions. All fly crosses were performed at 25˚C following standard genetic procedure unless otherwise specified.

For RNAi-mediated knockdown of target genes in oocytes, virgin females of the matalpha4-GAL-VP16 line were crossed with males of UAS-dsRNA lines, and their female progenies of right genotypes were collected and aged over yeast paste overnight, their egg chambers dissected for immunofluorescent staining. Phenotypes were evaluated based on yolk granule size, observed by DIC imaging, and also using a panel of markers, including Yl-eGFP for Yolk endocytosis and receptor recycling, PLCδ1(PH)-GFP for early endocytosis, 2xFYVE-GFP for early and late endosomes, Rab7 for late endosomes, Rab11 for recycling endosomes, and F-Actin for actin cytoskeleton dynamics.

The following fly lines were from Bloomington Drosophila Stock Center (BDSC): $yl^{13}$ $v^{24}$/FM3 (#4320). matalpha4-GAL-VP16 (#7062). w*;UAS-GFP-myc-2xFYVE (#42712)

Endogenous tagging lines for Rab4, Rab5 and Rab11: $y^1$, $w^{1118}$; TI{TI} EYFP-Rab4 (#62542). $w^{1118}$; TI{TI} EYFP-Rab5 /CyO (#62543). $w^{1118}$; TI{TI} EYFP-Rab11 (#62549).

UAS- based transgenic lines for YFP-tagged WT, DN and CA Rab proteins:

Rab4-WT (#9767), Rab4-CA (# 23268), Rab4-DN (#9768), Rab5-WT (#24616), Rab5-CA (#9774), Rab5-DN (#9771), Rab7-WT (#23641), Rab7-CA (#24103), Rab7-DN (#9778), Rab11-WT (#50782), Rab11-CA (#23260), Rab11-DN (#9792). Control of dsRNA for RNAi of Firefly Luciferase (FBgn0014448) under UAS control in the VALIUM1 vector (#31603).

dsRNA transgenic lines for RNAi-mediated knockdown of the following target genes: sktl (#27715), HK (#28330), CG7800 (#28922), Rabex-5 (#29357), mtm (#31552), mtm (#38339), mtm (#57298), Pi3K59F (#36056), Pi3K59F (#33384), VPS15 (#57011), VPS15 (#34092), VPS15 (#35209), Ocrl (#34722), Tbc1d15-17 (#34859), CG41099/ Rabankyrin-5, RIPK4,

ANKK1 (#34883), TBCK (#35332), CG17471 / PIP4K (#35338), CG17471 / PIP4K (#35660), Vps45 (#38252), YKT6 v-SNARE CG1515 (#50937), YKT6 v-SNARE CG1515 (#38314), Hip1 (#38377), Tbc1d15-17 (#43409), Rab4 (#33757), Rab5 (#67877), Rab7 (#27051), Rab11 (#27730), Rabex-5 (#50573), RUFY1 (#51494), Rbpn-5 (#52996), TBCK (#57223), Rbsn-5 (#57459), RUFY1 (#60496), NADSYN1 (#62265), Ccz1 (#62889), Bulli (regulator of MON1-CCZ1) (#63531), Pi3K59F (#64011), CG17471 / PIP4K (#65891), Hip1 (#77315), Rbcn-3A (#34612), Vha36-3 (#65075), Vha26 (#38996), Vha55 (#40884), Vha100-1 (#57860), Vha100-2 (#64859), VhaAC45RP (#64549), VhaAC39-2 (#67809), Vha68-2 (#34582).

## Genome tagging of *yl*

*yl* genome tagging constructs with eGFP and three copies of human influenza hemagglutinin (HA) tags were engineered as the following. BacPac clone CH321-59C03 (BACPAC Resource Center (BPRC), Children's Hospital Oakland Research Institute in Oakland, California), which covers all the genome coding regions of *yl* was selected as starting template for the tagging constructs. A 4.3kb SpeI and ApaI fragment covering N-terminal one third of *yl* genome region and an 8.0kb ApaI and HpaI fragment containing the remaining part of *yl* genome region were cloned separately from BacPac CH321-59C03 into PHSX cloning vector. KpnI and AscI restriction enzyme recognition sites were introduced in the middle of a 550 bp intermediate fragment encoding the C-terminal ST-rich region of Yl protein, and subsequently cloned into the 8.0kb ApaI/HpaI fragment in the PHSX vector. Separately, a DNA sequence encoding eGFP and 3 copies of HA tags flanked by KpnI and AscI restriction enzyme recognition sites were engineered and amplified by PCR, and then cloned into the modified 8.0kb ApaI/HpaI fragment in PHSX vector using the KpnI and AscI restriction enzyme sites. The 4.3kb N-terminal SpeI/ApaI *yl* fragment and the modified 8.0kb Apa1-Hpa1 *yl* fragment with eGFP-3xHA insertion were subsequently ligated together and cloned into the pCaspeR4 transgenic vector to assembly the ~14kb genome-tagging construct encoding full-length Yl protein with an eGFP and 3xHA tags near the C-terminus of encoded Yl protein. After DNA sequencing verification of the cloned construct, the purified DNA for the pCaspeR4-Yl-eGFP-3xHA tagging construct was injected into $w^{1118}$ embryos together with pπ25.7wc helper plasmid, and transgenes were selected and established following standard protocol [111]. The expression of tagged transgenes was validated by Western blots on protein extracts and by immunofluorescent stating by anti-eGFP and anti-HA antibodies from adult transgenic female flies (Figs 1–4 and 6–9).

## Dissection and immunofluorescent staining and imaging of egg chambers

Female flies of proper genotypes, aged 3 to 5 days, were fattened over yeast paste overnight, then egg chamber was were dissected in Drosophila S2 media containing 15% fetal bovine serum, fixed in 1xPBS with 4% paraformaldehyde for 20 minutes, follow by brief rinse with 1xPBS twice, wash with 1xPBT (1xPBS, plus 0.3% Triton X-100) for four times, 5 mins each, and then block with 5% normal goat or donkey serum in 1xPBT for 30 minutes. Incubate with primary antibodies at proper dilution in 1X PBT overnight at 4°C. Next day, remove the primary antibodies, wash the samples 5 times with 1X PBT. Leave the samples for 5–10 min in 1X PBT solution in between each wash. Incubate the washed samples in appropriate secondary antibodies as indicated dilution at room temperatures for 2 hours or at 4C overnight. Rinse 3 times with 1X PBS. Wash 6 times with 1X PBT with 10 min incubation between each wash. Incubate in 1:10,000 dilution of 1mg/mL 4', 6'-diamidino-2-phenylindole (DAPI) in 1XPBT solution for 10 min. Rinse 3 times with 1X PBS. Transfer egg chambers to slide with anti-fade

mounting medium (H-1900, Vector Laboratory) for imaging by a confocal microscope (Leica TCS SP5; Leica Microsystems, Wetzlar, Germany).

Each experiments were repeated independently at least once. In each experiment, at least 5 adult females of appropriate genotypes were dissected for immunofluorescent staining and microscopy imaging, 20 or more stage 10 egg chambers were inspected under immunofluorescent microscope, and five or more independent egg chambers were imaged through Z-serial section by confocal microscope.

## Antibodies

Antibodies were from the following sources: rat anti-Yl and rabbit anti-Yolk antibodies were from Dr. Mahowald [48]. mouse anti-Rab7 (1:10, Developmental Studies Hybridoma Bank (DSHB)), mouse anti-Rab11 (1:200, BD Transduction Laboratories (#610657)), mouse anti-actin (1:10000, MAB1501, Chemicon, and #ab-6276, Abcam); mouse anti-αTubulin (1:10000, DM1A, Sigma); mouse anti-HA (12CA5, Roche); chicken anti-GFP (Aves. 1:10,000 for Western blot and 1:2,000 for immunofluorescent staining). Alexa Fluor 488 AffiniPure Donkey Anti-Chicken IgY (1:1000, 703-545-155) and Rhodamine Red AffiniPure Donkey Anti-Mouse IgG (1:1000, 715-295-151) from Jackson ImmunoResearch Inc. TRITC-Phalloidin (Phalloidin–Tetramethylrhodamine B isothiocyanate) (1:200, P1951, Sigma). Alexa Fluor 488-Phalloidin (1:200, A-12379), Alexa Fluor 594-Phalloidin (1:200, A-12381), Alexa Fluor 647-Phalloidin (1:200, A-30107), Alexa Fluor 680-(A-21076) were from Molecular Probes (Invitrogen). Alexa Fluor 680-(A-21076) and Alexa Fluor 800-(926–32212) conjugated secondary antibodies for immunoblotting (1:10,000) were from Molecular Probes (Invitrogen) and LI-COR, respectively.

## Lysotracker staining

LysoTracker Red DND-99 (L7528, ThermoFisher Scientific) was used to detect acidic compartment in egg chambers following the manufacturer's instruction. Briefly, egg chambers were incubated with 200 μL of the 1:100 dilution of 1mM LysoTracker working solution in DMSO (final at 10uM) per tube of ovarioles for 5 minutes, fixed by adding 200 μL of 4% paraformaldehyde, followed by a 10-minute agitation at room temperature. This was succeeded by three wash cycles using 400 μL 1xPBST to permeabilize the tissue, 10 minutes each wash. The stained egg chambers were then visualized under a confocal microscope.

## Western blotting

Standard 8% SDS-PAGE gels were used for separation of Yl-eGFP-3xHA protein. The boiled samples were separated on SDS-PAGE and transferred to nitrocellulose membranes from Millipore. After blocking with 5% nonfat milk in Tris-buffered saline with 0.1% Tween-20 for 1 hour, membranes were incubated with primary antibodies. Secondary antibodies conjugated with Alexa-800 or Alexa-680 (Invitrogen) were used and the signals were detected by the Odyssey Infrared Imaging System and quantified by Odyssey Application Software 3.0 or by densitometry of the digital images using ImageJ software (NIH).

## Supporting information

**S1 Fig. Controls for anti-GFP and anti-Rab11 antibody staining in wildtype egg chambers.** Confocal fluorescent microscopy and DIC imaging of stage 10 egg chambers with immunofluorescent staining by anti-GFP antibody (green) on (A-D) $w^{1118}$ wild type control or (E, F) YFP-Rab11 endogenous tagging flies, co-labeled with (A-D) phalloidin for F-Actin and (C-F)

mouse anti-Rab11 antibody, as annotated. (B, D and F) High magnification views of the regions highlighted in (A, C and E), respectively. Images are presented as overlaying in color or individual channels in gray, as annotated. Genotypes: (A-D) *w^1118^*; (E,F), *w^1118^*; TI{TI} EYF-P-Rab11 (BDSC #62549). The sizes of the scale bars as annotated.
(TIF)

**S2 Fig. Subcellular localization of 2xFYVE-GFP reporter in wildtype and VPS34/PI3K59F-RNAi oocytes.** Confocal fluorescent microscopy and DIC imaging of stage 10 egg chambers with oocyte-specific expression of 2xFYVE-GFP reporter from (A-C) wild type control or (D-F) oocyte co-expressing dsRNA again VPS34/PI3K59F, triple-labeled for GFP (green), Rab7 (red) and F-Actin (blue). Images are presented as overlaying in color or individual channels in gray, as annotated. Genotypes: (A-C). w*/ *w^1118^*; matalpha4-GAL-VP16 (BDSC #7062), UAS-GFP-myc-2xFYVE (BDSC #42712)/+; (D-F). w*/ *w^1118^*; matalpha4-GAL-VP16 (BDSC #7062), UAS-GFP-myc-2xFYVE (BDSC #42712)/+; P{TRiP.HMJ30324} attP40 (BDSC #64011)/+. The sizes of the scale bars as annotated.
(TIF)

**S3 Fig. The roles of Rab5 on yolk granule biogenesis in vitellogenic oocytes.** Confocal fluorescent microscopy and DIC imaging of stage 10 egg chambers from oocytes of (A-D) wildtype control or (E-N) overexpressing (E-H) dominant-negative YFP-Rab5-DN, (I-L) constitutive-active YFP-Rab5-CA and (M, N) wildtype YFP-Rab5, that were (A, B, E, F, I, J, M, N) triple-labeled for GFP (green), endogenous Rab7 (red) and phalloidin for F-Actin (blue), or (C, D, G, H, K, L) stained with lysotracker alone (red), with data presented as overlaying images in color or individual channels in gray, as annotated. (B, F, J and N) High-magnification views of the cortex regions highlighted in (A, E, I and M), respectively, as annotated. The sizes of the scale bars as annotated. Genotypes: The samples were from adult female flies heterozygous for both matalpha4-GAL-VP16 (#7062) driver and the following UAS-transgenic lines: (A-D) *w^1118^*. (E-H) Rab5-DN: P{UASp-YFP.Rab5.S43N}01 (#9771). (I-L). Rab5-CA: P{UASp-YFP.Rab5.Q88L} (#9774). (M, N). Rab5-WT: P{UASp-YFP.Rab5}02 (#24616).
(TIF)

**S4 Fig. The roles of Rab11 on yolk granule biogenesis in vitellogenic oocytes.** Confocal fluorescent microscopy and DIC imaging of a stage 10 egg chambers with oocyte-specific expression of (A, B) constitutive active YFP-Rab11-CA or (C, D) wildtype YFP-Rab11 triple-labeled for YFP (green), endogenous Rab7 (red) and phalloidin for F-Actin (blue), presented as overlaying image of all channels in color or individual channels in gray, as annotated. (B, D) High-magnification views of the cortex regions highlighted in (A, C), respectively, as annotated. Genotypes: The samples were from adult female flies heterozygous for both matalpha4-GAL-VP16 driver (#7062) and the following UAS-transgenic lines: (A, B) Rab11-CA: P{UASp-YFP.Rab11.Q70L} (#23260). (C, D). Rab11-WT: P{UASp-YFP.Rab11} (#50782). The sizes of the scale bars as annotated.
(TIF)

**S5 Fig. The roles of Rab7 on yolk granule biogenesis in vitellogenic oocytes.** Confocal fluorescent microscopy and DIC imaging of stage 10 egg chambers with ectopic expression of (A-D) wildtype YFP-Rab7 or (E-H) constitutive active (CA) YFP-Rab7-CA in oocytes that are (A-F) co-labeled for anti-GFP (green), F-Actin and (A, B) anti-Rab7 or (C, D) lysotracker (red), or (G, H) by lysotracker (red) alone, shown as overlaying images of all the channels in color or in individual channels in gray, as annotated. (B, D, F) High-magnification view of the cortex regions highlighted in (A, C, E), respectively, as annotated. Genotypes: The samples were from adult female flies heterozygous for both matalpha4-GAL-VP16 driver (#7062) and

(A-D) wildtype YFP-Rab7: y[1 w*]; P{w(+mC) = UASp-YFP.Rab7}21/SM5 (#23641): (E, F) constitutive active YFP-Rab7-CA: P{UASp-YFP.Rab7.Q67L} (#24103).The sizes of the scale bars as annotated.
(TIF)

**S6 Fig. The roles of Rab4 on yolk granule biogenesis in vitellogenic oocytes.** Confocal fluorescent microscopy and DIC imaging of stage 10 egg chambers with oocyte-specific expression of (A, B) wildtype YFP-Rab4 or (C-F) constitutive active YFP-Rab4-CA (A-D) triple-labeled for YFP (green), endogenous Rab7 (red) and F-Actin (blue), shown in overlaying images in color or individual channels in gray, or (E, F) by lysotracker staining (red) alone, as annotated. (B, D) High-magnification views of the cortex regions highlighted in (A, C), respectively, as annotated. Genotypes: The samples were from adult female flies heterozygous for both matalpha4-GAL-VP16 driver (#7062) and the following UAS-transgenic lines: (A, B) wildtype Rab4: P{UASp-YFP.Rab4} (#9767). (C-F) Rab4-CA: P{UASp-YFP.Rab4.Q67L} (#9770). The sizes of the scale bars as annotated.
(TIF)

**S7 Fig. Essential roles of VPS34/VPS15 PI3 kinase complex in Yl recycling and yolk granule biogenesis.** Confocal fluorescent microscopy and DIC imaging of stage 10 egg chambers from flies carrying genome-tagging Yl-eGFP-3xHA with oocyte-specific expression of (A-C, G, H) control firefly Luciferase RNAi or (D-F, I, J) dsRNA again VPS34/PI3K59F, co-labeled with antibodies against GFP (green) and (A-F) lysotracker (red), or (G-J) endogenous Rab7 (red) and F-Actin (blue), as annotated. (B, E, H, J) High-magnification view of the cortex regions highlighted in (A, D, G, I), respectively. (C, F) Zoom-in view of the areas highlighted in (B, E), respectively. Images are presented as overlaying images in color or as individual channels in gray, as annotated. Genotypes: The samples were from adult females flies heterozygous for Yl-eGFP-3xHA reporter (p{mini-W+, *yl*-eGFP-3xHA}) and matalpha4-GAL-VP16 (BDSC #7062) driver together with (A-C, G, H). P{TRiP.JF01355}attP2 (BDSC#31603) or (D-F, I, J) P{TRiP.HMJ30324}attP40 (BDSC #64011). The sizes of the scales as annotated inside images.
(TIF)

**S8 Fig. Variable knockdown efficiency by different RNAi lines in stage 10 oocytes.** Confocal fluorescent microscopy and DIC imaging of stage 10 egg chambers with oocyte-specific expression of dsRNA against (A, B) Rab5, (C, D) Rab4 and (E, F) Rab7, co-stained for endogenous Rab7 (green) and F-Actin (red), as annotated. (B, D, F) High-magnification view of the cortex regions highlighted in (A, C, E), respectively. Images are presented as overlaying images in color or as individual channels in gray, as annotated. Genotypes: The samples were from adult females flies heterozygous for matalpha4-GAL-VP16 (BDSC #7062) driver together with (A, B) P{TRiP.GL01872}attP40 (BL# 67877). (C, D) P{TRiP.HMS01100}attP2P (BDSC #33757). (E,F) P{y(+t7.7] v(+t1.8) = TRiP.JF02377}attP2 (BDSC#27051). The sizes of the scales as annotated inside images.
(TIF)

**S1 Table. Summary of the pilot RNAi screen.**
(XLSX)

## Acknowledgments

We are grateful to Bloomington Drosophila Stock Center for *Drosophila* lines and Developmental Studies Hybridoma Bank (DSHB) and contributor Dr. Sean Munro for mouse anti-

Rab7 antibody[60] used in the study. Dr. Anthony Mahowald for rat anti-Yl and rabbit anti-Yolk antibodies. Dr. Zhenmei Mao at IMM Microscopy Core of UTHealth for confocal microscopy and imaging analyses.

## Author Contributions

**Conceptualization:** Sheng Zhang.

**Data curation:** Yue Yu, Dongsheng Chen, Stephen M. Farmer, Shiyu Xu, Beatriz Rios, Amanda Solbach, Xin Ye, Lili Ye.

**Formal analysis:** Yue Yu, Dongsheng Chen, Stephen M. Farmer, Shiyu Xu, Sheng Zhang.

**Funding acquisition:** Sheng Zhang.

**Investigation:** Dongsheng Chen, Sheng Zhang.

**Methodology:** Yue Yu, Dongsheng Chen, Sheng Zhang.

**Project administration:** Sheng Zhang.

**Resources:** Sheng Zhang.

**Supervision:** Sheng Zhang.

**Validation:** Yue Yu, Dongsheng Chen, Stephen M. Farmer, Shiyu Xu, Amanda Solbach, Xin Ye.

**Visualization:** Yue Yu, Sheng Zhang.

**Writing – original draft:** Sheng Zhang.

**Writing – review & editing:** Yue Yu, Sheng Zhang.

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
