## [Decision Letter · Decision Letter 0]

12 Sep 2023

Dear Dr  Zhang,

Thank you very much for submitting your Research Article entitled 'Endolysosomal trafficking controls yolk granule biogenesis in vitellogenic Drosophila oocytes' to PLOS Genetics.

The manuscript was fully evaluated at the editorial level and by independent peer reviewers. The reviewers appreciated the attention to an important problem, but raised some substantial concerns about the current manuscript. Based on the reviews, we will not be able to accept this version of the manuscript, but we would be willing to review a revised version. We cannot, of course, promise publication at that time.

If you decide to revise the manuscript for further consideration at PLOS Genetics, please aim to resubmit within the next 60 days, unless it will take extra time to address the concerns of the reviewers, in which case we would appreciate an expected resubmission date by email to plosgenetics@plos.org.

We are sorry that we cannot be more positive about your manuscript at this stage. Please do not hesitate to contact us if you have any concerns or questions.

Yours sincerely,

Pablo Wappner

Academic Editor

PLOS Genetics

Gregory P. Copenhaver

Editor-in-Chief

PLOS Genetics

Reviewer's Responses to Questions

**Comments to the Authors:**

Reviewer #1: In this manuscript, Yu et al. shed light on the significance of Rab5, Rab11, Rab7, dynein motor, and the RAVE/V-ATPase complex in endocytosis, endosomal trafficking, and yolk granule biogenesis. The authors generated a tagged version of Yolkless (YI) and studied its distribution in vitellogenic oocytes. They observed a connection between different endosomal structures, phosphoinositides, and actin cytoskeleton dynamics. The study identified Rab5 and Rab11 as crucial players in yolk granule biogenesis, while Rab4 and Rab7 role was dispensable. The authors carried out a small RNAi screen on Rab5 effector molecules identified in a previous proteomic study and proposed the involvement of the dynein motor and the RAV/V-ATPase complexes in the early stages of endosomal trafficking.

The data presented in this manuscript offers novel insights into endolysosomal pathway, and highlight the vitellogenic oocyte in Drosophila as a valuable in vivo system to study endocytosis and endolysosomal trafficking. Such insights are of broad interest to researchers studying complex membrane trafficking events.

This is an exciting and clearly-written manuscript. The results are well organized, and the data support the interpretation and conclusions drawn. However, some revisions are recommended to improve the manuscript.

1. To assess the functionality of the transgenic construct Yl-eGFP-3xHA, was it placed in yl mutant background?

2. For readability, it would be beneficial to combine Figure-1 and Figure-2 (Inserting Figure-2 panel between Fig.1D and Fig.1E)

3. In Fig.1G, please provide an inset of invaginating clathrin-coated pits/vesicles if available.

4. Please cite this article “Crystalline yolk spheroids in Drosophila melanogaster oocyte: Freeze fracture and two-dimensional reconstruction analysis” (https://doi.org/10.1016/j.jinsphys.2006.12.011) when referring to the condensed and crystalized yolk protein in mature granules.

5. Briefly describe how the transgenic construct Yl-eGFP-3xHA was used to rescue the sterility of yl mutant females.

6. It would be beneficial to place an arrow in the figures when referring to F-actin puncta (E.g. Fig.4B and 4I)

7. What were the selection criteria for selecting specific Rab5 effectors for the RNAi screens?

8. RNAi lines might have a variable knockdown efficiency and potential off-targets, since most of the tested lines did not show any effect on granule biogenesis in the RNAi screen, was qPCR performed to the test the knockdown efficiency?

9. It would be great if the authors can clarify the number of replicates performed to validate the observed phenotype in the RNAi screens, either in the figure legend or the methods section.

10. The study focuses on stage 10 oocytes, and the findings might be specific to this developmental stage. Independent of the context and potential redundancy it would help readers to discuss the general applicability of the findings in other cell types, tissues, or organisms.

Reviewer #2: In their manuscript Yu and colleagues are showing a detailed microscopic analysis of the endocytic pathway that is essential for the generation of the yolk granules in the Drosophila oocyte. The authors use multiple Rab transgenes and Rab5 effector RNAis to genetically dissect the sequence of the events of this pathway. Although the study is comprehensive and aims to give a more complete view about this phenomenon, it still has multiple flaws, like it relies almost entirely on one technique (confocal microscopy) but lacking any quantifications and contains multiple formal mistakes. Thus, the manuscript requires substantial improvement before it may get considered for acceptance in this Journal.

Major comments:

1. The first and most important flaw is the complete lack of quantifications and statistics. As the novelty factor of the findings is moderate the comprehensiveness could the major advantage of the study. However, it is really hard to compare the effect of the different genetic modifications without any quantifications. For example, visually the phenotype upon loss of of Rbcn3 may seems similar to the silencing of the class III PI3K complex, but it is really hard to make any conclusions without statistics.

2. The authors are using the overexpression of DN forms of the Rab proteins to show their loss of function phenotype. However, RabDN-s are frequently cause weaker loss of function phenotype than mutant alleles or RNAi-s. So, it would be important to confirm the effect of DN lines by silencing of Rab4, Rab5, Rab7 and Rab11 by appropriate RNAi lines.

3. In the data from the small-scale screen on Rab5 effectors (Table S1) it is visible that in some cases the used RNAi lines that silence the same gene (like in the case of Vps15) showed different phenotypes. Was the effectivity of RNAi lines tested (like with qPCR)?

4. Authors sometimes claim interesting suggestions based on fluorescent mictoscopy data, but these are remaining as speculations without testing them by molecular methods. For example, based on the results with ctp RNAi Authors suggest that the dynein-dynactin complex would work as the effector of Rab5 instead of Rab7 in this tissue. Although this is an attractive idea it should be tested by pull down or immunoprecipitations.

5. Upon the knockdown of PI3K59F (Fig6C) it is clearly visible that remarkable amount of FYVE-GFP positive structures. Which is surprising, as loss of PI3K59F should diminish the formation of PI3P positive granules. Was the RNAi efficient enough? What can be the identity of these persistent FYVE-GFP granules? Are these persistent FYVE granules are colocalizing with the remaining Rab7 puncta?

6. Fig6A shows that upon the KD of PI3K59 YL-GFP is leaving the cortical region and delivered further inside into the oocyte. Does YL-GFP is reaching the lysosomes upon these conditions? Although the overlap between the diffuse signal of YL-GFP and Rab7 is apparent, but there are still some Rab7 and YL-GFP puncta remaining in these cells. Are these YL-GFP puncta colocalizing with Rab7 or lysosomal markers?

7. On Fig9 the Authors are using the silencing of two Vha genes to show the effect of loss of the V-Atpase on yolk granule formation. However, they are using different RNAi lines to show the effect on Rab7 (Vha26 RNAi), and FYVE-GFP (Vha68 RNAi). Although the phenotypes are consistent, to rule out any concern of pleiotropic effects it would be better to use the same RNAi for both markers (or use both RNAis for both).

Minor comments:

1. The Legends for main figures and supplementary figures and sometimes the figures itself contains multiple inconsistence and annoying mistakes:

- Fig1: What are the green circles that appear on panels E, F, G? In the legend the second reference for panel (F) actually refers to panel (G)

- Fig4: The UASp-PLCD1(PH) reporter is called as PLCD-eGFP in the legend, but PH-GFP is written on the image panel, this inconsistence should be resolved. The legend refers to panel (G) that shows FYVE-GFP, however on the image this panel shows PH-GFP

- FigS3-S5: The logic of the panel labels of FigS3-S5 is different from the logic that was followed by the rest of the Figures. This inconsistence should be resolved.

- FigS5: The order of panel labels restarts after the 10th panel (panel J followed by another A, B, C, etc.) and inconsistent with the legend. This should be corrected.

2. On the 18th page of the submitted PDF Authors are writing the following: “overexpressing either DN-, WT- or CA-YFP-Rab7, as large numbers of lysotracker-positive yolk granules were present in these oocytes (Fig 5C-F and Fig. S4)”, however there are no panel showing the YFP-Rab7WT on these figures.

3. There is no reference for Fig7E, F panels in the main text.

Reviewer #3: Endocytic recycling is a fundamental cellular process that plays a key role in all tissues, and yet it has mostly been studied in mammalian tissue culture cells or yeast. The Drosophila model system has a lot of potential as a means to investigate how endocytic processes vary between tissues and are regulated to meet their disparate needs.

The Rab GTPases are known to be master regulators of the endocytic pathway, and this paper reports an analysis of the role of key Rabs in the endocytosis of vitellogenin and yolk granule formation in developing oocytes. These maturing oocytes are very large and so amenable to imaging, and also accessible with tools such as ectopic protein expression and RNAi for following cellular processes and perturbing protein expression. The authors apply these methods to investigate the contributions that Rabs 4, 5, 7 and 11 make to the endocytosis and recycling of Yolkless (the vitellogenin receptor) and the formation of yolk granules. They also investigate the effect of knocking down a set of c 16 putative Rab5 effectors that were identified in a previous study based on affinity chromatography. Overall, the authors conclude that Rab5 and Rab11 play key roles in the process and identify several Rab5 effectors that are particularly important. They also find that Rab7 is not critical for Yolkless recycling and granule formation which is an unexpected finding but convincingly shown by both the Rab7 reagents and the knock-down of a Rab7 GEF. This raises interesting questions about what purpose Rab7 serves in such recycling pathways in tissues.

The paper describes a lot of work, and the data consists almost entirely of a large number of immunofluorescence images which are of a good quality and are clearly presented with both lower magnification overviews and higher magnification images to show details. The text is clear, although it would benefit from condensing and also some grammatical tidying. Overall, the paper makes a useful contribution to the endocytosis field as it shows the potential of the oocyte system, and in particular its value for studying Rab5 and Rab11. However, before publication the following relatively minor issues would need to be addressed:

a) The text is rather long in the Introduction, Results and Discussion. Being more succinct would encourage more people to read the paper. It would be good to get the input from someone with lots of paper writing experience.

b) For most of the figures the single IMF channels are shown as grey scale which makes them easier to see than colour. However, this is not the case for Figure 3, and so this should be altered to follow the better practice of the other figures.

c) The authors state in the Introduction that PI3P acts as a landmark to enlist Ccz1/Mon1, but as they state later it is actually Rab5 that does this. There are other PI3P binders that could be mentioned instead.

d) For figures that show a YFP-Rab, the authors should include the YPF in the labelling on the figure and not just the name of the Rab to make clear that it is a tagged protein (ie YFP-Rab5 rather than Rab5).

e) Did the authors look at the distribution of Rab11 in an oocyte that is not also expressing YFP-Rab11? If so, it should be shown.

f) In Figure S2 they could include a wild-type control showing Rab7 and actin so as to aid comparisons. Perhaps the figure could be split into two figures to facilitate this.

g) Figure 5D is labelled Rab-DN but it should be Rab7-DN.

h) The authors should mention that flies lacking Rab4 are viable and fertile (in contrast to the situation for the other Rabs that they investigate). See PubMed ID 33666175.

i) The section on cut up (ctp) could be deleted as it is not directly linked to the Rab work in the rest of the paper.

k) It would be helpful to have a summary table showing the effect on the various markers of all of the different Rab perturbing treatments tested.

**Have all data underlying the figures and results presented in the manuscript been provided?**

Reviewer #1: Yes

Reviewer #2: Yes

Reviewer #3: Yes

PLOS authors have the option to publish the peer review history of their article (what does this mean?). If published, this will include your full peer review and any attached files.

Reviewer #1: No

Reviewer #2: No

Reviewer #3: No

---

## [Decision Letter · Decision Letter 1]

16 Jan 2024

Dear Dr Zhang,

Thank you very much for submitting the revised version of your Article entitled 'Endolysosomal trafficking controls yolk granule biogenesis in vitellogenic Drosophila oocytes' to PLOS Genetics.

The manuscript was fully evaluated at the editorial level and by the same reviewers that evaluated the original version. The reviewers feel that all their concerns have been properly addressed, although they raised a few  minor points that you need to change before the manuscript can be accepted for publication

We therefore ask you to modify the manuscript according to the review recommendations. Your revisions should address the specific points made by each reviewer.

Yours sincerely,

Pablo Wappner

Academic Editor

PLOS Genetics

Gregory P. Copenhaver

Editor-in-Chief

PLOS Genetics

Reviewer's Responses to Questions

**Comments to the Authors:**

Reviewer #1: The authors have done an excellent job of addressing and responding to all the comments and concerns raised during the review process. They have diligently offered well-reasoned explanations and additional data where necessary to support their findings and conclusions.

Reviewer #2: The Authors addressed most of my concerns experimentally. I also approve their arguments regarding the difficulties of the quantification. This problem was further addressed by the use of a large number of samples (20< oocytes/genotype). Overall, the data and the manuscript's quality greatly improved.

I still have one minor comment: The texts "Yl" and (DIC) on Fig1 G1 and G2 have been mixed up. This should be corrected.

Reviewer #3: In revising this manuscript the authors have done an excellent job of addressing the points that I made. It was already a good quality study and I am now very happy to recommend that it be published. I have one minor suggestion which is that the newly added Table S1 seems a valuable summary of all the data which would help readers. Thus it might be good to incorporate it into the main paper rather that include it as a supplementary Excel file.

**Have all data underlying the figures and results presented in the manuscript been provided?**

Reviewer #1: Yes

Reviewer #2: Yes

Reviewer #3: Yes

PLOS authors have the option to publish the peer review history of their article (what does this mean?). If published, this will include your full peer review and any attached files.

Reviewer #1: No

Reviewer #2: No

Reviewer #3: No

---

## [Editor Report · Decision Letter 2]

22 Jan 2024

Dear Dr Zhang,

We are pleased to inform you that your manuscript entitled "Endolysosomal trafficking controls yolk granule biogenesis in vitellogenic Drosophila oocytes" has been editorially accepted for publication in PLOS Genetics. Congratulations!

Yours sincerely,

Pablo Wappner

Academic Editor

PLOS Genetics

Gregory P. Copenhaver

Editor-in-Chief

PLOS Genetics

Comments from the reviewers (if applicable):

**Data Deposition**

http://datadryad.org/submit?journalID=pgenetics&manu=PGENETICS-D-23-00857R2

**Press Queries**

---

## [Editor Report · Acceptance letter]

29 Jan 2024

PGENETICS-D-23-00857R2 

Endolysosomal trafficking controls yolk granule biogenesis in vitellogenic Drosophila oocytes 

Dear Dr Zhang, 

We are pleased to inform you that your manuscript entitled "Endolysosomal trafficking controls yolk granule biogenesis in vitellogenic Drosophila oocytes" has been formally accepted for publication in PLOS Genetics! Your manuscript is now with our production department and you will be notified of the publication date in due course.

With kind regards,

Zsofia Freund

PLOS Genetics

On behalf of:
